# Group-Relative REINFORCE is Secretly an Off-Policy Algorithm: Demystifying Some Myths About GRPO and its Friends

**Chaorui Yao** [*]
University of California, Los Angeles
chaorui@ucla.edu

**Yanxi Chen** [*]
Alibaba Group
chenyanxi.cyx@alibaba-inc.com

**Yuchang Sun, Yushuo Chen, Wenhao Zhang**
Alibaba Group
{sunyuchang.syc,chenyushuo.cys,zwh434786}@alibaba-inc.com

**Xuchen Pan, Yaliang Li, Bolin Ding**
Alibaba Group
{panxuchen.pxc,yaliang.li,bolin.ding}@alibaba-inc.com

## Abstract

Off-policy reinforcement learning (RL) for large language models (LLMs) is attracting growing interest, driven by practical constraints in real-world applications, the complexity of LLM-RL infrastructure, and the need for further innovations of RL methodologies. While classic REINFORCE and its modern variants like Group Relative Policy Optimization (GRPO) are typically regarded as on-policy algorithms with limited tolerance of off-policyness, we present in this work a first-principles derivation for *group-relative REINFORCE* — a REINFORCE variant that uses the within-group mean reward as the baseline for advantage calculation — without assuming a specific training data distribution, showing that it admits a *native off-policy interpretation*. This perspective yields two general principles for adapting REINFORCE to truly off-policy settings: regularizing policy updates, and actively shaping the data distribution. Our analysis demystifies some myths about the roles of importance sampling and clipping in GRPO, unifies and reinterprets two recent algorithms — Online Policy Mirror Descent and Asymmetric REINFORCE — as regularized forms of the REINFORCE loss, and offers theoretical justification for seemingly heuristic data-weighting strategies. Our findings lead to actionable insights that are validated with extensive empirical studies, and open up new opportunities for principled algorithm design in off-policy RL for LLMs. Source code for this work is available at https://github.com/agentscope-ai/Trinity-RFT/tree/main/examples/rec_gsm8k.

## 1 Introduction

The past few years have witnessed rapid progress in reinforcement learning (RL) for large language models (LLMs). This began with reinforcement learning from human feedback (RLHF) (Bai et al., 2022; Ouyang et al., 2022) that aligns pre-trained LLMs with human preferences, followed by reasoning-oriented RL that enables LLMs to produce long chains of thought (OpenAI, 2024; DeepSeek-AI, 2025; Kimi-Team, 2025b; Zhang et al., 2025b). More recently, agentic RL (Kimi-Team, 2025a; Gao et al., 2025; Zhang et al., 2025a) aims to train LLMs for agentic capabilities such as tool use, long-horizon planning, and multi-step task execution in dynamic environments.

---

[*]Equal contribution. Part of the work was done while Chaorui Yao was an intern at Alibaba Group.

Alongside these developments, off-policy RL has been attracting growing interest. In the "era of experience" (Silver & Sutton, 2025), LLM-powered agents need to be continually updated through interaction with the environment. Practical constraints in real-world deployment and the complexity of LLM-RL infrastructure often render on-policy training impractical (Noukhovitch et al., 2025): rollout generation and model training can proceed at mismatched speeds, data might be collected from different policies, reward feedback might be irregular or delayed, and the environment may be too costly or unstable to query for fresh trajectories. Moreover, in pursuit of higher sample efficiency and model performance, it is desirable to go beyond the standard paradigm of independent rollout sampling, e.g., via replaying past experiences (Schaul et al., 2016; Rolnick et al., 2019; An et al., 2025), synthesizing higher-quality experiences based on auxiliary information (Da et al., 2025; Liang et al., 2025; Guo et al., 2025), or incorporating expert demonstrations into online RL (Yan et al., 2025; Zhang et al., 2025c) — all of which incur off-policyness.

However, the prominent algorithms in LLM-RL — Proximal Policy Optimization (PPO) (Schulman et al., 2017) and Group Relative Policy Optimization (GRPO) (Shao et al., 2024) — are essentially on-policy methods: as modern variants of REINFORCE (Williams, 1992), their fundamental rationale is to produce unbiased estimates of the policy gradient, which requires fresh data sampled from the current policy. PPO and GRPO can handle a limited degree of off-policyness via importance sampling, but require that the current policy remains sufficiently close to the behavior policy. Truly off-policy LLM-RL often demands ad-hoc analysis and algorithm design; worse still, as existing RL infrastructure (Sheng et al., 2024; Hu et al., 2024; von Werra et al., 2020; Wang et al., 2025b; Pan et al., 2025; Fu et al., 2025a) is typically optimized for REINFORCE-style algorithms, their support for specialized off-policy RL algorithms could be limited. All these have motivated our investigation into principled and infrastructure-friendly algorithm design for off-policy RL.

**Core finding: a native off-policy interpretation for group-relative REINFORCE.** Consider a one-step RL setting and a group-relative variant of REINFORCE that, like in GRPO, assumes access to multiple responses $\{y_1, \ldots, y_K\}$ for the same prompt $x$ and use the group mean reward $\bar{r}$ as the baseline in advantage calculation. Each response is a sequence of tokens $y_i = (y_i^1, y_i^2, \ldots)$, and receives a response-level reward $r_i = r(x, y_i)$. Let $\pi_{\boldsymbol{\theta}}(\cdot \mid x)$ denote an autoregressive policy parameterized by $\boldsymbol{\theta}$. The update rule for each iteration of group-relative REINFORCE is $\boldsymbol{\theta}' = \boldsymbol{\theta} + \eta \boldsymbol{g}$, where $\eta$ is the learning rate, and $\boldsymbol{g}$ is the sum of updates from multiple prompts and their corresponding responses. For a specific prompt $x$, the update would be[1]

$$\boldsymbol{g}\big(\boldsymbol{\theta}; x, \{y_i, r_i\}_{1 \leq i \leq K}\big) = \frac{1}{K} \sum_{1 \leq i \leq K} (r_i - \bar{r}) \nabla_{\boldsymbol{\theta}} \log \pi_{\boldsymbol{\theta}}(y_i \mid x) \qquad \text{(response-wise)} \quad \text{(1a)}$$

$$= \frac{1}{K} \sum_{1 \leq i \leq K} \sum_{1 \leq t \leq |y_i|} (r_i - \bar{r}) \nabla_{\boldsymbol{\theta}} \log \pi_{\boldsymbol{\theta}}(y_i^t \mid x, y_i^{<t}) \quad \text{(token-wise)} \quad \text{(1b)}$$

Here, the response-wise and token-wise formulas are linked by the elementary decomposition $\log \pi_{\boldsymbol{\theta}}(y_i \mid x) = \sum_t \log \pi_{\boldsymbol{\theta}}(y_i^t \mid x, y_i^{<t})$, where $y_i^{<t}$ denotes the first $t-1$ tokens of $y_i$.

A major finding of this work is that group-relative REINFORCE admits a native off-policy interpretation. We establish this in Section 2 via a novel, first-principles derivation that makes no explicit assumption about the sampling distribution of the responses $\{y_i\}$, in contrast to the standard policy gradient theory. Our derivation provides a new perspective for understanding how REINFORCE makes its way towards the optimal policy by constructing a series of surrogate objectives and taking gradient steps for the corresponding surrogate losses. Such analysis can be extended to multi-step RL settings as well, with details deferred to Appendix A.

**Implications: principles and concrete methods for augmenting REINFORCE.** While the proposed off-policy interpretation does not imply that vanilla REINFORCE should converge to

---

[1]For notational simplicity and consistency, we use the same normalization factor $1/K$ for both response-wise and token-wise formulas in Eq. (1a) and (1b). For practical implementation, the gradient is calculated with samples from a mini-batch, and typically normalized by the total number of response tokens. This mismatch does not affect our theoretical studies in this work. Interestingly, our analysis of REINFORCE in this work provides certain justifications for calculating the token-mean loss within a mini-batch, instead of first taking the token-mean loss within each sequence and then taking the average across sequences (Shao et al., 2024); our perspective is complementary to the rationales explained in prior works like DAPO (Yu et al., 2025), although a deeper understanding of this aspect is beyond our current focus.

the optimal policy when given arbitrary training data (which is too good to be true), our analysis in Section 3 identifies two general principles for augmenting REINFORCE in off-policy settings: (1) regularize the policy update step to stabilize learning, and (2) actively shape the training data distribution to steer the policy update direction. As we will see in Section 4, this unified framework demystifies common myths about the rationales behind many recent RL algorithms: (1) It reveals that in GRPO, clipping (as a form of regularization) plays a much more essential role than importance sampling, and it is often viable to enlarge the clipping range far beyond conventional choices for accelerated convergence without sacrificing stability. (2) Two recent algorithms — Kimi's Online Policy Mirror Descent (OPMD) (Kimi-Team, 2025b) and Meta's Asymmetric REINFORCE (AsymRE) (Arnal et al., 2025) — can be reinterpreted as adding a regularization loss to the standard REINFORCE loss, which differs substantially from the rationales explained in their original papers. (3) Our framework justifies heuristic data-weighting strategies like discarding certain low-reward samples or up-weighting high-reward ones, even though they violate assumptions in policy gradient theory and often require ad-hoc analysis in prior works.

Extensive empirical studies in Section 4 and Appendix B validate these insights and demonstrate the efficacy and/or limitations of various algorithms under investigation. By revealing the off-policy nature of group-relative REINFORCE, our work opens up new opportunities for principled, infrastructure-friendly algorithm design in off-policy LLM-RL with solid theoretical foundation.

## 2 TWO INTERPRETATIONS FOR REINFORCE

Consider the standard reward-maximization objective in reinforcement learning:

$$\max_{\boldsymbol{\theta}} \quad J(\boldsymbol{\theta}) \coloneqq \mathbb{E}_{x \sim D} \, J(\boldsymbol{\theta}; x), \quad \text{where} \quad J(\boldsymbol{\theta}; x) \coloneqq \mathbb{E}_{y \sim \pi_{\boldsymbol{\theta}}(\cdot|x)} \, r(x, y), \tag{2}$$

where $D$ is a distribution over the prompts $x$.

We first recall the standard on-policy interpretation of REINFORCE in Section 2.1, and then present our proposed off-policy interpretation in Section 2.2.

### 2.1 RECAP: ON-POLICY INTERPRETATION VIA POLICY GRADIENT THEORY

In the classical on-policy view, REINFORCE updates policy parameters $\boldsymbol{\theta}$ using samples that are drawn directly from $\pi_{\boldsymbol{\theta}}$. The policy gradient theorem (Sutton et al., 1998) tells us that

$$\nabla_{\boldsymbol{\theta}} J(\boldsymbol{\theta}; x) = \nabla_{\boldsymbol{\theta}} \, \mathbb{E}_{y \sim \pi_{\boldsymbol{\theta}}(\cdot|x)} \, r(x, y) = \mathbb{E}_{y \sim \pi_{\boldsymbol{\theta}}(\cdot|x)} \Big[ \big( r(x, y) - b(x) \big) \, \nabla_{\boldsymbol{\theta}} \log \pi_{\boldsymbol{\theta}}(y|x) \Big],$$

where $b(x)$ is a baseline for reducing variance when $\nabla_{\boldsymbol{\theta}} J(\boldsymbol{\theta}; x)$ is estimated with finite samples. If samples are drawn from a different behavior policy $\pi_{\mathsf{b}}$ instead, the gradient can be rewritten as

$$\nabla_{\boldsymbol{\theta}} J(\boldsymbol{\theta}; x) = \mathbb{E}_{y \sim \pi_{\mathsf{b}}(\cdot|x)} \Big[ \big( r(x, y) - b(x) \big) \, \frac{\pi_{\boldsymbol{\theta}}(y \mid x)}{\pi_{\mathsf{b}}(y \mid x)} \, \nabla_{\boldsymbol{\theta}} \log \pi_{\boldsymbol{\theta}}(y \mid x) \Big].$$

While the raw importance-sampling weight $\pi_{\boldsymbol{\theta}}(y|x)/\pi_b(y|x)$ facilitates unbiased policy gradient estimate, it may be unstable when $\pi_{\boldsymbol{\theta}}$ and $\pi_{\mathsf{b}}$ diverge. Modern variants of REINFORCE address this by modifying the probability ratios (e.g., via clipping or normalization), which achieves better bias-variance trade-off in the policy gradient estimate and leads to a stable learning process.

In the LLM context, we have $\nabla_{\boldsymbol{\theta}} \log \pi_{\boldsymbol{\theta}}(y \mid x) = \sum_t \nabla_{\boldsymbol{\theta}} \log \pi_{\boldsymbol{\theta}}(y^t \mid x, y^{<t})$, but the response-wise probability ratio $\pi_{\boldsymbol{\theta}}(y|x)/\pi_{\mathsf{b}}(y|x)$ can blow up or shrink exponentially with the sequence length. Practical implementations typically adopt token-wise probability ratio instead:

$$\widetilde{g}(\boldsymbol{\theta}; x) = \mathbb{E}_{y \sim \pi_{\mathsf{b}}(\cdot|x)} \Big[ \big( r(x, y) - b(x) \big) \sum_{1 \leq t \leq |y|} \frac{\pi_{\boldsymbol{\theta}}(y^t \mid x, y^{<t})}{\pi_{\mathsf{b}}(y^t \mid x, y^{<t})} \, \nabla_{\boldsymbol{\theta}} \log \pi_{\boldsymbol{\theta}}(y^t \mid x, y^{<t}) \Big]$$

Although this becomes a biased approximation of $\nabla_{\boldsymbol{\theta}} J(\boldsymbol{\theta}; x)$, classical RL theory still offers policy improvement guarantees if $\pi_{\boldsymbol{\theta}}$ is sufficiently close to $\pi_{\mathsf{b}}$ (Kakade & Langford, 2002; Fragkiadaki, 2018; Schulman et al., 2015; 2017; Achiam et al., 2017).

## 2.2 A NEW OFF-POLICY INTERPRETATION FOR GROUP-RELATIVE REINFORCE

We now provide an alternative off-policy interpretation for group-relative REINFORCE. Let us think of policy optimization as an iterative process $\boldsymbol{\theta}_1, \boldsymbol{\theta}_2, \ldots$, and focus on the $t$-th iteration that updates the policy model parameters from $\boldsymbol{\theta}_t$ to $\boldsymbol{\theta}_{t+1}$. Our derivation consists of three steps: (1) define a KL-regularized surrogate objective, and show that its optimal solution must satisfy certain consistency conditions; (2) define a surrogate loss (with finite samples) that enforces such consistency conditions; and (3) take one gradient step of the surrogate loss, which turns out to be equivalently the group-relative REINFORCE method.

**Step 1: surrogate objective and consistency condition.** Consider the following KL-regularized surrogate objective that incentivizes the policy to make a stable improvement over $\pi_{\boldsymbol{\theta}_t}$:

$$\max_{\boldsymbol{\theta}} \quad J(\boldsymbol{\theta}; \pi_{\boldsymbol{\theta}_t}) := \mathbb{E}_{x \sim D}\Big[\mathbb{E}_{y \sim \pi_{\boldsymbol{\theta}}(\cdot|x)}[r(x,y)] - \tau \cdot D_{\mathsf{KL}}\big(\pi_{\boldsymbol{\theta}}(\cdot|x) \,\|\, \pi_{\boldsymbol{\theta}_t}(\cdot|x)\big)\Big], \tag{3}$$

where $\tau$ is a regularization coefficient. It is a well-known fact that the optimal policy $\pi$ for this surrogate objective satisfies the following (Ziebart et al., 2008) (Nachum et al., 2017; Korbak et al., 2022; Rafailov et al., 2023; Richemond et al., 2024; Kimi-Team, 2025b): for any prompt $x$ and response $y$,

$$\pi(y|x) = \frac{\pi_{\boldsymbol{\theta}_t}(y|x)e^{r(x,y)/\tau}}{Z(x, \pi_{\boldsymbol{\theta}_t})}, \quad \text{where } Z(x, \pi_{\boldsymbol{\theta}_t}) := \int \pi_{\boldsymbol{\theta}_t}(y'|x)e^{r(x,y')/\tau} \, \mathrm{d}y'. \tag{4}$$

Note that Eq. (4) is equivalent to the following: for any pair of responses $y_1$ and $y_2$,

$$\frac{\pi(y_1|x)}{\pi(y_2|x)} = \frac{\pi_{\boldsymbol{\theta}_t}(y_1|x)}{\pi_{\boldsymbol{\theta}_t}(y_2|x)} \exp\left(\frac{r(x,y_1) - r(x,y_2)}{\tau}\right).$$

Taking logarithm of both sides, we have this *pairwise consistency condition*:

$$r_1 - \tau \cdot \big(\log \pi(y_1|x) - \log \pi_{\boldsymbol{\theta}_t}(y_1|x)\big) = r_2 - \tau \cdot \big(\log \pi(y_2|x) - \log \pi_{\boldsymbol{\theta}_t}(y_2|x)\big). \tag{5}$$

**Step 2: surrogate loss with finite samples.** Given a prompt $x$ and $K$ responses $y_1, \ldots, y_K$, we define the following mean-squared surrogate loss that enforces the consistency condition, as done in prior works (Gao et al., 2024; Flet-Berliac et al., 2024):

$$\widehat{L}(\boldsymbol{\theta}; x, \pi_{\boldsymbol{\theta}_t}) := \frac{1}{K^2} \sum_{1 \leq i < j \leq K} \frac{(a_i - a_j)^2}{(1+\tau)^2}, \quad \text{where } a_i := r_i - \tau\Big(\log \pi_{\boldsymbol{\theta}}(y_i|x) - \log \pi_{\boldsymbol{\theta}_t}(y_i|x)\Big). \tag{6}$$

Here, we normalize $a_i - a_j$ by $1 + \tau$ to account for the loss scale. In theory, if this surrogate loss is defined by infinite samples with sufficient coverage of the action space (Song et al., 2024), then its unique minimizer is the same as the optimal policy for the surrogate objective in Eq. (3).

**Step 3: one gradient step of the surrogate loss.** Let us conduct further analysis for $(a_i - a_j)^2$. The trick here is that, if we take only one gradient step of this loss at $\boldsymbol{\theta} = \boldsymbol{\theta}_t$, then the values of $\log \pi_{\boldsymbol{\theta}}(y_i|x) - \log \pi_{\boldsymbol{\theta}_t}(y_i|x)$ and $\log \pi_{\boldsymbol{\theta}}(y_j|x) - \log \pi_{\boldsymbol{\theta}_t}(y_j|x)$ are simply zero. As a result,

$$\nabla_{\boldsymbol{\theta}}(a_i - a_j)^2\big|_{\boldsymbol{\theta}_t} = -2\tau(r_i - r_j)\Big(\nabla_{\boldsymbol{\theta}} \log \pi_{\boldsymbol{\theta}}(y_i|x)\big|_{\boldsymbol{\theta}_t} - \nabla_{\boldsymbol{\theta}} \log \pi_{\boldsymbol{\theta}}(y_j|x)\big|_{\boldsymbol{\theta}_t}\Big) \quad \Rightarrow$$

$$\nabla_{\boldsymbol{\theta}} \sum_{1 \leq i < j \leq K} \frac{(a_i - a_j)^2}{(1+\tau)^2}\bigg|_{\boldsymbol{\theta}_t} = \sum_{i < j} \frac{-2\tau}{(1+\tau)^2}(r_i - r_j)\Big(\nabla_{\boldsymbol{\theta}} \log \pi_{\boldsymbol{\theta}}(y_i|x)\big|_{\boldsymbol{\theta}_t} - \nabla_{\boldsymbol{\theta}} \log \pi_{\boldsymbol{\theta}}(y_j|x)\big|_{\boldsymbol{\theta}_t}\Big)$$

$$= \sum_{i < j} \frac{-2\tau}{(1+\tau)^2}\Big((r_i - r_j)\nabla_{\boldsymbol{\theta}} \log \pi_{\boldsymbol{\theta}}(y_i|x)\big|_{\boldsymbol{\theta}_t} + (r_j - r_i)\nabla_{\boldsymbol{\theta}} \log \pi_{\boldsymbol{\theta}}(y_j|x)\big|_{\boldsymbol{\theta}_t}\Big)$$

$$= \frac{-2\tau}{(1+\tau)^2} \sum_{1 \leq i \leq K} \sum_{1 \leq j \leq K} (r_i - r_j)\nabla_{\boldsymbol{\theta}} \log \pi_{\boldsymbol{\theta}}(y_i|x)\big|_{\boldsymbol{\theta}_t}$$

$$= \frac{-2\tau K}{(1+\tau)^2} \sum_{1 \leq i \leq K} (r_i - \bar{r})\nabla_{\boldsymbol{\theta}} \log \pi_{\boldsymbol{\theta}}(y_i|x)\big|_{\boldsymbol{\theta}_t}, \quad \text{where} \quad \bar{r} := \frac{1}{K} \sum_{1 \leq j \leq K} r_j.$$

Putting these back to the surrogate loss defined in Eq. (6), we end up with this policy update step:

$$g\big(\boldsymbol{\theta}; x, \{y_i, r_i\}_{1 \le i \le K}\big) = \frac{2\tau}{(1+\tau)^2} \cdot \frac{1}{K} \sum_{1 \le i \le K} \left(r_i - \bar{r}\right) \nabla_{\boldsymbol{\theta}} \log \pi_{\boldsymbol{\theta}}(y_i \,|\, x). \tag{7}$$

That's it! We have just derived the group-relative REINFORCE method, but without any on-policy assumption about the distribution of training data $\{x, \{y_i, r_i\}_{1 \le i \le K}\}$. The regularization coefficient $\tau > 0$ controls the update step size; a larger $\tau$ effectively corresponds to a smaller learning rate.

**Summary and remarks.** Figure 1 visualizes the proposed interpretation of what REINFORCE is actually doing. The curve going through $\boldsymbol{\theta}_t \rightarrow \boldsymbol{\theta}_{t+1} \rightarrow \widetilde{\boldsymbol{\theta}}_{t+1} \rightarrow \boldsymbol{\theta}^\star$ stands for the ideal optimization trajectory from $\boldsymbol{\theta}_t$ to the optimal policy model $\boldsymbol{\theta}^\star$, if the algorithm solves each intermediate surrogate objective $J(\boldsymbol{\theta}; \pi_{\boldsymbol{\theta}_t})$ / surrogate loss $\widehat{L}(\boldsymbol{\theta}; \pi_{\boldsymbol{\theta}_t})$ exactly at each iteration $t$. In comparison, REINFORCE is effectively taking a single gradient step of the surrogate loss and immediately moving on to the next iteration $\boldsymbol{\theta}_{t+1}$ with a new surrogate objective.

Two remarks are in place. (1) Our derivation of group-relative REINFORCE can be generalized to multi-step RL settings, by replacing a response $y$ in the previous analysis with a full trajectory consisting of multiple turns of agent-environment interaction. For example, regarding the surrogate objective in Eq. (3), we need to replace the response-level reward and KL divergence with their trajectory-level counterparts. Interested readers might refer to Appendix A for the full analysis. (2) The above analysis suggests that we might interpret group-relative REINFORCE from a *pointwise or pairwise* perspective. While the policy update in Eq. (7) is stated in a pointwise manner, we have also seen that, at each iteration, REINFORCE is implicitly enforcing the pairwise consistency condition in Eq. (5) among multiple responses. This allows us the flexibility to choose whichever perspective that offers more intuition for our analysis later in this work.

## 3    PITFALLS AND AUGMENTATIONS

Although we have provided a native off-policy interpretation for REINFORCE, it certainly does not guarantee convergence to the optimal policy when given arbitrary training data. This section identifies pitfalls that could undermine vanilla REINFORCE, which motivate two principles for augmentations in off-policy settings.

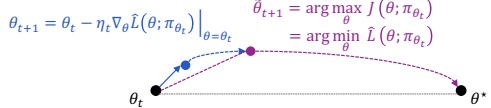

Figure 1: A visualization of our off-policy interpretation for group-relative REINFORCE. Here $\widehat{L}(\boldsymbol{\theta}; \pi_{\boldsymbol{\theta}_t}) = \mathbb{E}_{x \sim \widehat{D}}[\widehat{L}(\boldsymbol{\theta}; x, \pi_{\boldsymbol{\theta}_t})]$, where $\widehat{D}$ is the sampling distribution for prompts and $\widehat{L}(\boldsymbol{\theta}; x, \pi_{\boldsymbol{\theta}_t})$ is the loss defined in Eq. (6).

**Pitfalls of vanilla REINFORCE.** In Figure 1, we might expect that ideally, (1) $\widetilde{\boldsymbol{\theta}}_{t+1} - \boldsymbol{\theta}_t$ aligns with the direction of $\boldsymbol{\theta}^\star - \boldsymbol{\theta}_t$; and (2) $\boldsymbol{\theta}_{t+1} - \boldsymbol{\theta}_t$ aligns with the direction of $\widetilde{\boldsymbol{\theta}}_{t+1} - \boldsymbol{\theta}_t$. One pitfall, however, is that even if both conditions hold, they do *not* necessarily imply that $\boldsymbol{\theta}_{t+1} - \boldsymbol{\theta}_t$ should align well with $\boldsymbol{\theta}^\star - \boldsymbol{\theta}_t$. That is, $\langle \widetilde{\boldsymbol{\theta}}_{t+1} - \boldsymbol{\theta}_t, \boldsymbol{\theta}^\star - \boldsymbol{\theta}_t \rangle > 0$ and $\langle \boldsymbol{\theta}_{t+1} - \boldsymbol{\theta}_t, \widetilde{\boldsymbol{\theta}}_{t+1} - \boldsymbol{\theta}_t \rangle > 0$ do not imply $\langle \boldsymbol{\theta}_{t+1} - \boldsymbol{\theta}_t, \boldsymbol{\theta}^\star - \boldsymbol{\theta}_t \rangle > 0$. Moreover, it is possible that $\boldsymbol{\theta}_{t+1} - \boldsymbol{\theta}_t$ might not align well with $\widetilde{\boldsymbol{\theta}}_{t+1} - \boldsymbol{\theta}_t$. Recall from Eq. (7) that, from $\boldsymbol{\theta}_t$ to $\boldsymbol{\theta}_{t+1}$, we take one gradient step for a surrogate loss that enforces the pairwise consistency condition among a *finite* number of samples. Given the enormous action space of an LLM, some implicit assumptions about the training data (e.g., balancedness and coverage) would be needed to ensure that the gradient aligns well with the direction towards the optimum of the surrogate objective, namely $\widetilde{\boldsymbol{\theta}}_{t+1} - \boldsymbol{\theta}_t$.

In fact, without a mechanism that ensures boundedness of policy update under a sub-optimal data distribution, vanilla REINFORCE could eventually converge to a sub-optimal policy. Let us show this with a minimal example in a didactic 3-arm bandit setting. Suppose that there are three actions $\{a_j\}_{1 \le j \le 3}$ with rewards $\{r(a_j)\}$. Consider $K$ training samples $\{y_i\}_{1 \le i \le K}$, where $y_i \in \{a_j\}_{1 \le j \le 3}$ is sampled from some behavior policy $\pi_{\mathsf{b}}$. Denote by $\mu_r := \sum_{1 \le j \le 3} \pi_{\mathsf{b}}(a_j) r(a_j)$ the expected reward under $\pi_{\mathsf{b}}$, and $\bar{r} := \sum_i r(y_i)/K$ the average reward of training samples. We consider the softmax parameterization, i.e., $\pi_{\boldsymbol{\theta}}(a_j) = e^{\theta_j} / \sum_\ell e^{\theta_\ell}$ for a policy parameterized by $\boldsymbol{\theta} \in \mathbb{R}^3$. A standard fact is that $\nabla_{\boldsymbol{\theta}} \log \pi_{\boldsymbol{\theta}}(a_j) = \boldsymbol{e}_j - \pi_{\boldsymbol{\theta}}$, where $\boldsymbol{e}_j \in \mathbb{R}^3$ is a one-hot vector with value 1 at

entry $j$. Now we examine the policy update direction of REINFORCE, as $K \to \infty$:

$$\boldsymbol{g} = \frac{1}{K} \sum_{1 \le i \le K} (r(y_i) - \bar{r}) \nabla_{\boldsymbol{\theta}} \log \pi_{\boldsymbol{\theta}}(y_i) \to \sum_{1 \le j \le 3} \pi_{\mathsf{b}}(a_j)(r(a_j) - \mu_r) \nabla_{\boldsymbol{\theta}} \log \pi_{\boldsymbol{\theta}}(a_j)$$

$$= \sum_{1 \le j \le 3} \pi_{\mathsf{b}}(a_j)(r(a_j) - \mu_r)(\boldsymbol{e}_j - \pi_{\boldsymbol{\theta}}) = \sum_{1 \le j \le 3} \pi_{\mathsf{b}}(a_j)(r(a_j) - \mu_r)\boldsymbol{e}_j.$$

For example, if $\boldsymbol{r} = [r(a_j)]_{1 \le j \le 3} = [0, 0.8, 1]$ and $\pi_{\mathsf{b}} = [0.3, 0.6, 0.1]$, then basic calculation says $\mu_r = 0.58$, $\boldsymbol{r} - \mu_r = [-0.58, 0.22, 0.42]$, and finally $g_2 = \pi_{\mathsf{b}}(a_2)(r(a_2) - \mu_r) > \pi_{\mathsf{b}}(a_3)(r(a_3) - \mu_r) = g_3$, which implies that the policy will converge to the sub-optimal action $a_2$.

**Two principles for augmenting REINFORCE.**    The identified pitfalls of vanilla REINFORCE suggest two general principles for augmenting REINFORCE in off-policy scenarios:

- One is to *regularize the policy update step*, ensuring that the optimization trajectory remains bounded and reasonably stable when given training data from a sub-optimal distribution;

- The other is to *steer the policy update direction*, by actively weighting the training samples rather than naively using them as is.

These two principles are not mutually exclusive, and might be integrated within a single algorithm. We will see in the next section that many RL algorithms can be viewed as instantiations of them.

## 4    RETHINKING THE RATIONALES BEHIND RECENT RL ALGORITHMS

This section revisits various RL algorithms through a unified lens — the native off-policy interpretation of group-relative REINFORCE and its augmentations — and demystifies some common myths about their working mechanisms. Our main findings are summarized as follows:

| ID | Finding | Analysis & Experiments |
|---|---|---|
| F1 | GRPO's effectiveness in off-policy settings stems from *clipping as regularization* rather than importance sampling. A wider clipping range than usual often accelerates training without harming stability. | Section 4.1, Figures 2, 3, 5, 8, 9 |
| F2 | Kimi's OPMD and Meta's AsymRE can be interpreted as *REINFORCE loss + regularization loss*, complementary to the rationales in their original papers. | Section 4.2, Figure 10 |
| F3 | *Data-oriented heuristics* — such as dropping excess negatives or up-weighting high-reward rollouts — fit naturally into our off-policy view and show strong empirical performance. | Section 4.3, Figures 4, 5, 11 |

**Experimental setup.**    We conduct experiments with the Trinity-RFT framework (Pan et al., 2025), and control off-policyness with the `sync_interval` (frequency of model synchronization) and `sync_offset` (lag between rollout generation and training) parameters. Larger values of these parameters improve efficiency (via pipeline parallelism) at the cost of off-policyness; in addition, `sync_offset > 0` simulates delayed environmental feedback in practical scenarios. We also include a stress-test setting that only allows access to offline data generated by the initial policy model. Our experiments cover math reasoning tasks like GSM8k (Cobbe et al., 2021), MATH (Hendrycks et al., 2021), Guru-Math (Cheng et al., 2025), and tool-use tasks like ToolACE (Liu et al., 2025a). LLMs under consideration include Qwen2.5-1.5B-Instruct, Qwen2.5-7B-Instruct (Qwen-Team, 2025a), Llama-3.1-8B-Instruct, Llama-3.2-3B-Instruct (Dubey et al., 2024), and Qwen3-30B-A3B (Qwen-Team, 2025b). Further details can be found in Appendix B.

### 4.1    DEMYSTIFYING MYTHS ABOUT GRPO

Recall that in GRPO, the advantage for each response $y_i$ is defined as $A_i = (r_i - \bar{r})/\sigma_r$, where $\bar{r}$ and $\sigma_r$ denote the within-group mean and standard deviation of the rewards $\{r_i\}_{1 \le i \le K}$ respectively.

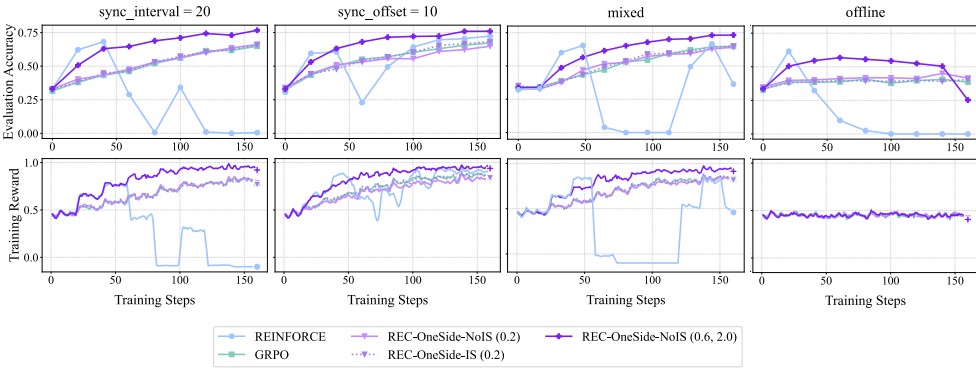

Figure 2: Empirical results for REC algorithms on GSM8k with Qwen2.5-1.5B-Instruct. Training reward curves are smoothed with a running-average window of size 3. Numbers in the legend denote clipping parameters $\epsilon_{\mathsf{low}}, \epsilon_{\mathsf{high}}$. The "mixed" setting adopts `sync_interval = 16` and `sync_offset = 8`.

We consider the practical implementation of GRPO with token-wise importance-sampling (IS) weighting and clipping, whose loss function for a specific prompt $x$ and responses $\{y_i\}$ is[2]

$$\widehat{L} = \frac{1}{K} \sum_{1 \leq i \leq K} \sum_{1 \leq t \leq |y_i|} \min\left\{ \frac{\pi_{\boldsymbol{\theta}}(y_i^t \,|\, x, y_i^{<t})}{\pi_{\mathsf{old}}(y_i^t \,|\, x, y_i^{<t})} A_i, \ \mathrm{clip}\left( \frac{\pi_{\boldsymbol{\theta}}(y_i^t \,|\, x, y_i^{<t})}{\pi_{\mathsf{old}}(y_i^t \,|\, x, y_i^{<t})}, 1 - \epsilon_{\mathsf{low}}, 1 + \epsilon_{\mathsf{high}} \right) A_i \right\},$$

where $\pi_{\mathsf{old}}$ denotes the older policy version that generated this group of rollout data. The gradient of this loss can be written as (Schulman et al., 2017)

$$\boldsymbol{g}\big(\boldsymbol{\theta}; x, \{y_i, r_i\}_{1 \leq i \leq K}\big) = \frac{1}{K} \sum_{1 \leq i \leq K} \sum_{1 \leq t \leq |y_i|} \nabla_{\boldsymbol{\theta}} \log \pi_{\boldsymbol{\theta}}(y_i^t \,|\, x, y_i^{<t}) \cdot A_i \frac{\pi_{\boldsymbol{\theta}}(y_i^t \,|\, x, y_i^{<t})}{\pi_{\mathsf{old}}(y_i^t \,|\, x, y_i^{<t})} M_i^t,$$

where $M_i^t$ denotes a one-side clipping mask:

$$M_i^t = \mathbb{1}\left( A_i > 0, \ \frac{\pi_{\boldsymbol{\theta}}(y_i^t \,|\, x, y_i^{<t})}{\pi_{\mathsf{old}}(y_i^t \,|\, x, y_i^{<t})} \leq 1 + \epsilon_{\mathsf{high}} \right) + \mathbb{1}\left( A_i < 0, \ \frac{\pi_{\boldsymbol{\theta}}(y_i^t \,|\, x, y_i^{<t})}{\pi_{\mathsf{old}}(y_i^t \,|\, x, y_i^{<t})} \geq 1 - \epsilon_{\mathsf{low}} \right). \quad (8)$$

**Ablation study with the REC series.** To isolate the roles of importance sampling and clipping, we consider a series of REINFORCE-with-Clipping (REC) algorithms. Due to space limitation, we defer our studies of more clipping mechanisms to Appendix B.3, and focus on REC with one-side clipping in this section. More specifically, REC-ONESIDE-IS removes advantage normalization in GRPO (to reduce variability), and REC-ONESIDE-NOIS further removes IS weighting:

$$\text{REC-ONESIDE-IS: } \boldsymbol{g} = \frac{1}{K} \sum_{1 \leq i \leq K} \sum_{1 \leq t \leq |y_i|} \nabla_{\boldsymbol{\theta}} \log \pi_{\boldsymbol{\theta}}(y_i^t \,|\, x, y_i^{<t}) \cdot (r_i - \overline{r}) \frac{\pi_{\boldsymbol{\theta}}(y_i^t \,|\, x, y_i^{<t})}{\pi_{\mathsf{old}}(y_i^t \,|\, x, y_i^{<t})} M_i^t,$$

$$\text{REC-ONESIDE-NOIS: } \boldsymbol{g} = \frac{1}{K} \sum_{1 \leq i \leq K} \sum_{1 \leq t \leq |y_i|} \nabla_{\boldsymbol{\theta}} \log \pi_{\boldsymbol{\theta}}(y_i^t \,|\, x, y_i^{<t}) \cdot (r_i - \overline{r}) M_i^t.$$

**Experiments.** We conduct experiments to validate Finding F1 regarding the roles of clipping (with a small or large clipping range) and importance sampling in GRPO. Figure 2 presents GSM8k results with Qwen2.5-1.5B-Instruct in various off-policy settings. REC-ONESIDE-IS/NOIS and GRPO (with the same $\epsilon_{\mathsf{low}} = \epsilon_{\mathsf{high}} = 0.2$) have nearly identical performance, indicating that importance sampling is non-essential, whereas the collapse of REINFORCE highlights the critical role of clipping. Radically enlarging $(\epsilon_{\mathsf{low}}, \epsilon_{\mathsf{high}})$ to $(0.6, 2.0)$ accelerates REC-ONESIDE-NOIS

---

[2]In our experiments with GRPO, we neglect KL regularization with respect to an extra reference model, or entropy regularization that encourages output diversity. Recent works (Yu et al., 2025; Liu et al., 2025b) have shown that these practical techniques are often unnecessary.

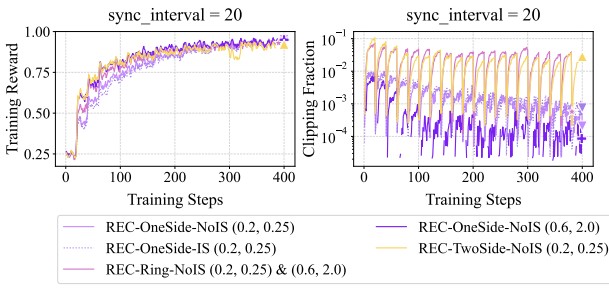

Figure 3: Empirical results for REC on ToolACE with Llama-3.2-3B-Instruct. Training reward curves are smoothed with a running-average window of size 3. Details about REC-TwoSide and REC-Ring are provided in Appendix B.3.

without compromising stability in all considered settings (except "offline"). Similar patterns also appear in Figure 3 (ToolAce with Llama-3.2-3B-Instruct) and other results in Appendix B. As for the stress-test ("offline") setting, Figure 2 reveals an intrinsic trade-off between learning speed and stability, motivating future work toward better algorithms that achieve both. We hypothesize that sequence-level importance sampling in GSPO (Zheng et al., 2025) could be non-essential as well; interested readers may refer to our preliminary results in Appendix B.7 that support this prediction.

## 4.2 UNDERSTANDING KIMI'S OPMD AND META'S ASYMRE

Besides clipping, another natural method is to add a regularization loss $R(\cdot)$ to vanilla REINFORCE:

$$\widehat{L}\big(\boldsymbol{\theta}; x, \{y_i, r_i\}_{1 \leq i \leq K}\big) = -\frac{1}{K} \sum_{i \in [K]} (r_i - \overline{r}) \log \pi_{\boldsymbol{\theta}}(y_i \,|\, x) + \beta \cdot R\big(\boldsymbol{\theta}; x, \{y_i, r_i\}_{1 \leq i \leq K}\big),$$

and take $\boldsymbol{g} = -\nabla_{\boldsymbol{\theta}}\widehat{L}$. We show below that Kimi's OPMD and Meta's AsymRE are indeed special cases of this unified formula, with empirical validation of their efficacy deferred to Appendix B.5.

**Kimi's OPMD.** Kimi-Team (2025b) derives an OPMD variant by taking logarithm of both sides of Eq. (4), which leads to a consistency condition and further motivates the following surrogate loss:

$$\widetilde{L} = \frac{1}{K} \sum_{1 \leq i \leq K} \left( r_i - \tau \log Z(x, \pi_{\boldsymbol{\theta}_t}) - \tau \left( \log \pi_{\boldsymbol{\theta}}(y_i \,|\, x) - \log \pi_{\boldsymbol{\theta}_t}(y_i | x) \right) \right)^2.$$

With $K$ responses generated by $\pi_{\mathsf{old}} = \pi_{\boldsymbol{\theta}_t}$, the term $\tau \log Z(x, \pi_{\boldsymbol{\theta}_t})$ can be *approximated* by a finite-sample estimate $\tau \log(\sum_i e^{r_i/\tau}/K)$ (Brantley et al., 2025), which can be further *approximated* by the mean reward $\overline{r} = \sum_i r_i/K$ if $\tau$ is large. With these approximations, the gradient of $\widetilde{L}$ becomes equivalent to that of the following loss (which is the final version of Kimi's OPMD):

$$\widehat{L} = -\frac{1}{K} \sum_{1 \leq i \leq K} (r_i - \overline{r}) \log \pi_{\boldsymbol{\theta}}(y_i \,|\, x) + \frac{\beta}{2K} \sum_{1 \leq i \leq K} \left( \log \pi_{\boldsymbol{\theta}}(y_i \,|\, x) - \log \pi_{\mathsf{old}}(y_i \,|\, x) \right)^2, \text{ where } \beta = \tau.$$

In comparison, our analysis in Sections 2 and 3 suggests that this is in itself a principled loss function for off-policy RL, adding a mean-squared regularization loss to the vanilla REINFORCE loss.

**Meta's AsymRE.** AsymRE (Arnal et al., 2025) modifies REINFORCE by tuning down the baseline (from $\overline{r}$ to $\overline{r} - \beta$) in advantage calculation, which was motivated by the intuition of prioritizing learning from positive samples and justified by multi-arm bandit analysis in the original paper. We offer an alternative interpretation for AsymRE by rewriting its loss function:

$$\widehat{L} = -\frac{1}{K} \sum_i \left( r_i - (\overline{r} - \beta) \right) \log \pi_{\boldsymbol{\theta}}(y_i \,|\, x) = -\frac{1}{K} \sum_i (r_i - \overline{r}) \log \pi_{\boldsymbol{\theta}}(y_i \,|\, x) - \frac{\beta}{K} \sum_i \log \pi_{\boldsymbol{\theta}}(y_i \,|\, x).$$

Note that the first term on the right-hand side is the REINFORCE loss, and the second term serves as regularization, enforcing imitation of responses from an older version of the policy model. For the latter, we may also add a term that is independent of $\boldsymbol{\theta}$ to it and take the limit $K \to \infty$:

$$-\frac{1}{K} \sum_{1 \leq i \leq K} \log \pi_{\boldsymbol{\theta}}(y_i \,|\, x) + \frac{1}{K} \sum_{1 \leq i \leq K} \log \pi_{\mathsf{old}}(y_i \,|\, x) = \frac{1}{K} \sum_{1 \leq i \leq K} \log \frac{\pi_{\mathsf{old}}(y_i \,|\, x)}{\pi_{\boldsymbol{\theta}}(y_i \,|\, x)}$$

$$\to \mathbb{E}_{y\sim\pi_{\mathsf{old}}(\cdot\,|\,x)}\left[\log\frac{\pi_{\mathsf{old}}(y\,|\,x)}{\pi_{\boldsymbol{\theta}}(y\,|\,x)}\right] = D_{\mathsf{KL}}\Big(\pi_{\mathsf{old}}(\cdot|x)\|\pi_{\boldsymbol{\theta}}(\cdot|x)\Big),$$

which turns out to be a finite-sample approximation of KL regularization.

### 4.3 UNDERSTANDING DATA-WEIGHTING METHODS

We now shift our attention to the second principle for augmenting REINFORCE, i.e., actively shaping the training data distribution.

**Pairwise weighting.** Recall from Section 2 that we define the surrogate loss in Eq. (6) as an unweighted sum of pairwise mean-squared losses. However, if we have certain knowledge about which pairs are more informative for RL training, we may assign higher weights to them. This motivates generalizing $\sum_{i<j}(a_i - a_j)^2$ to $\sum_{i<j} w_{i,j}(a_i - a_j)^2$, where $\{w_{i,j}\}$ are non-negative weights. Assuming that $w_{i,j} = w_{j,i}$ and following the steps in Section 2, we end up with

$$\boldsymbol{g}\big(\boldsymbol{\theta}; x, \{y_i, r_i\}_{1\leq i\leq K}\big) = \frac{1}{K}\sum_{1\leq i\leq K}\Big(\sum_{1\leq j\leq K} w_{i,j}\Big)\Big(r_i - \frac{\sum_j w_{i,j} r_j}{\sum_j w_{i,j}}\Big)\nabla_{\boldsymbol{\theta}}\log\pi_{\boldsymbol{\theta}}(y_i\,|\,x).$$

In the special case where $w_{i,j} = w_i w_j$, this becomes

$$\boldsymbol{g} = \Big(\sum_j w_j\Big)\frac{1}{K}\sum_{1\leq i\leq K} w_i\big(r_i - \overline{r}_w\big)\nabla_{\boldsymbol{\theta}}\log\pi_{\boldsymbol{\theta}}(y_i\,|\,x), \ \ \text{where } \overline{r}_w := \frac{\sum_j w_j r_j}{\sum_j w_j}. \tag{9}$$

Based on this, we investigate two REINFORCE-with-data-weighting (RED) methods.

**RED-DROP: sample dropping.** The idea is to use a filtered subset $\mathcal{S} \subseteq [K]$ of responses for training (Shang et al., 2025). For example, the Kimi-Researcher blog (Kimi-Team, 2025a) proposes to "discard some negative samples strategically", as negative gradients increase the risk of entropy collapse. This is indeed a special case of Eq. (9), by setting $w_i = \sqrt{K}/|\mathcal{S}|$ for $i \in \mathcal{S}$ and 0 otherwise:

$$\boldsymbol{g}\big(\boldsymbol{\theta}; x, \{y_i, r_i\}_{1\leq i\leq K}\big) = \frac{1}{|\mathcal{S}|}\sum_{i\in\mathcal{S}}(r_i - \overline{r}_{\mathcal{S}})\nabla_{\boldsymbol{\theta}}\log\pi_{\boldsymbol{\theta}}(y_i\,|\,x), \ \ \text{where } \overline{r}_{\mathcal{S}} = \frac{1}{|\mathcal{S}|}\sum_{i\in\mathcal{S}} r_i. \tag{10}$$

While this is no longer an unbiased estimate of policy gradient even if all responses are sampled from the current policy, it is still well justified by our off-policy interpretation of REINFORCE.

**RED-WEIGHT: pointwise loss weighting.** Another approach for prioritizing high-reward responses is to directly up-weight their gradient terms in Eq. (1a). To better understand the working mechanism of this seemingly heuristic method, we rewrite its policy update:

$$\boldsymbol{g} = \sum_{1\leq i\leq K} w_i(r_i - \overline{r})\nabla_{\boldsymbol{\theta}}\log\pi_{\boldsymbol{\theta}}(y_i|x) = \sum_{1\leq i\leq K} w_i(r_i - \overline{r}_w + \overline{r}_w - \overline{r})\nabla_{\boldsymbol{\theta}}\log\pi_{\boldsymbol{\theta}}(y_i|x)$$

$$= \sum_{1\leq i\leq K} w_i(r_i - \overline{r}_w)\nabla_{\boldsymbol{\theta}}\log\pi_{\boldsymbol{\theta}}(y_i|x) + (\overline{r}_w - \overline{r})\sum_{1\leq i\leq K} w_i\nabla_{\boldsymbol{\theta}}\log\pi_{\boldsymbol{\theta}}(y_i|x).$$

This is the pairwise-weighted REINFORCE gradient in Eq. (9), plus a regularization term (weighted by $\overline{r}_w - \overline{r} > 0$) that resembles the one in AsymRE but prioritizes imitating higher-reward responses, echoing the finding from offline RL literature (Hong et al., 2023a;b) that regularizing against high-reward trajectories can be more effective than conservatively imitating all trajectories in the dataset.

**Experiments.** Figure 4 presents GSM8k results with Qwen2.5-1.5B-Instruct, which confirm the efficacy of RED-DROP and RED-WEIGHT (details in Appendix B.6) in on/off-policy settings, comparable to REC-ONESIDE-NOIS with enlarged $(\epsilon_{\mathsf{low}}, \epsilon_{\mathsf{high}})$. Figure 5 reports larger-scale experiments on Guru-Math with Qwen2.5-7B-Instruct, where RED-WEIGHT achieves higher rewards than GRPO, with similar KL distance to the initial policy. Figure 11 in the appendix further validates the efficacy of RED-WEIGHT on MATH with Llama-3.1-8B-Instruct.

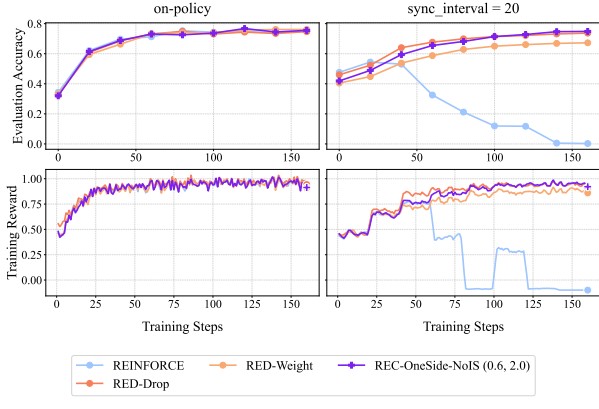

Figure 4: Empirical performance of RED algorithms on GSM8k with Qwen2.5-1.5B-Instruct, in both on-policy and off-policy settings. Training reward curves are smoothed with a running-average window of size 3. Implementation details about RED-WEIGHT and RED-DROP are provided in Appendix B.6.

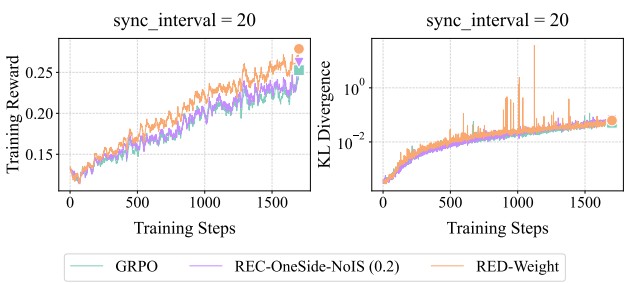

Figure 5: Empirical results on Guru-Math with Qwen2.5-7B-Instruct. Training reward curves are smoothed with a running-average window of size 3.

## 5 DISCUSSIONS

**Related works.** Off-policy RL for LLMs has been studied from various perspectives. Importance sampling has long been considered one foundational mechanism for off-policy RL; besides TRPO, PPO and GRPO, recent extensions include GSPO (Zheng et al., 2025) and GMPO (Zhao et al., 2025) that work with sequence-wise probability ratios, CISPO (Chen et al., 2025) that clips probability ratios rather than token updates, decoupled PPO (Fu et al., 2025a) that adapts PPO to asynchronous RL, among others (Roux et al., 2025; Zheng et al., 2026; Xi et al., 2026; Wang et al., 2025a). AsymRE (Arnal et al., 2025) offers an alternative baseline-shift approach (with ad-hoc analysis for discrete bandit settings), while OPMD (Kimi-Team, 2025b) partly overlaps with our analysis up to Eq. (4) before diverging, as discussed earlier in Section 4.2. REBEL (Gao et al., 2024) and CoPG (Flet-Berliac et al., 2024) overlap with our analysis up to Eq. (6) before diverging, which will be elaborated in Appendix D. Other perspectives for off-policy LLM-RL include learning dynamics of DPO and SFT (Ren & Sutherland, 2025), training offline loss functions with negative gradients on on-policy data (Tajwar et al., 2024), or improving generalization of SFT via probability-aware rescaling (Wu et al., 2025). Another line of research integrates expert data into online RL (Yan et al., 2025; Zhang et al., 2025c; Fu et al., 2025b). Our work contributes complementary perspectives to this growing toolkit for off-policy LLM-RL. Further discussion on prior works that are most closely related to our main analysis can be found in Appendix D.

**Limitations and future work.** While our work offers a new off-policy interpretation for group-relative REINFORCE and shows its broad implications for LLM-RL, several limitations remain. (1) Our current analysis covers single/multi-step RL with response/trajectory-level rewards, and assumes access to multiple rollouts per query. Future work may expand its scope and applicability, e.g., generalizing to settings with step-level rewards or only one rollout per query. (2) Our analysis lacks formal guarantees for policy improvement or convergence. Future work may identify distributional assumptions that yield provable guarantees for REINFORCE variants in off-policy settings. (3) Our experiments focus on settings where training data is generated by older policy versions. Extensions to broader off-policy settings, e.g., advanced experience synthesis (Shi et al., 2026) or incorporation of expert data (Yan et al., 2025; Zhang et al., 2025c), may reveal new insights. Addressing these limitations will further solidify the theoretical foundation and advance principled algorithm design for off-policy LLM-RL.

ACKNOWLEDGMENTS

The authors would like to thank the anonymous reviewers and Area Chairs for their constructive feedback that has helped improve this work.

REPRODUCIBILITY STATEMENT

Full implementation details and hyperparameter configurations are documented in Section 4 and Appendix B. We have released our code publicly to facilitate reproducibility.

ETHICS STATEMENT

All datasets used in this study (e.g., GSM8k, MATH, Guru, ToolACE) are publicly available, and no private or personally identifiable information was collected or used. Our contributions are methodological, focusing on improving the stability and efficiency of RL for LLM post-training. We acknowledge that LLMs may still generate biased or harmful outputs; however, our experiments are restricted to benchmark evaluations and do not involve deployment in real-world systems. We believe that releasing our code and reporting detailed hyperparameter settings will foster reproducibility and responsible advancement in this field.

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

## LLM USAGE STATEMENT

We used large language models (LLMs) only as general-purpose writing assistants to polish the presentation and improve the clarity of the text. All research contributions and findings are solely the work of the authors.

## A   EXTENDING SECTION 2.2 TO MULTI-STEP RL

This section extends the off-policy interpretation proposed in Section 2.2 to multi-step RL settings. Let us start by introducing some notations. In multi-step RL, the initial prompt $x$ is also regarded as the initial state $s^1 = x$. A rollout trajectory consisting of multiple turns of agent-environment interaction is denoted by

$$\mathcal{T} = (s^1, a^1, s^2, a^2, \dots) = (s^\ell, a^\ell)_{1 \le \ell \le |\mathcal{T}|},$$

where $s^\ell$ is the state and $a^\ell$ is the action, i.e., an LLM response (akin to $y$ in Section 2.2). Let $c^\ell$ denote the context up to step $\ell$, so that $a^\ell \sim \pi(\cdot|c^\ell)$ for some policy $\pi$. Throughout this section, we consider trajectory-level rewards $r(x, \mathcal{T})$. Let $\rho_{\boldsymbol{\theta}}(\cdot|x)$ denote the trajectory distribution induced by policy $\pi_{\boldsymbol{\theta}}$ at initial state $s^1 = x$.

The following analysis focuses on the $t$-th iteration, updating the policy model from $\boldsymbol{\theta}_t$ to $\boldsymbol{\theta}_{t+1}$.

**Step 1: surrogate objective and consistency condition.** For the $t$-th iteration of policy optimization, consider the following KL-regularized objective:

$$\max_{\boldsymbol{\theta}} \quad J(\boldsymbol{\theta}; \pi_{\boldsymbol{\theta}_t}) := \mathbb{E}_{x \sim D}\left[ \mathbb{E}_{\mathcal{T} \sim \rho_{\boldsymbol{\theta}}(\cdot|x)}[r(x, \mathcal{T})] - \tau \cdot D_{\mathsf{KL}}\big(\rho_{\boldsymbol{\theta}}(\cdot|x) \,\|\, \rho_{\boldsymbol{\theta}_t}(\cdot|x)\big) \right]. \tag{11}$$

The optimal policy $\pi$ and the induced trajectory distribution $\rho$ satisfies the following: for any trajectory $\mathcal{T}$,

$$\rho(\mathcal{T}|x) = \frac{\rho_{\boldsymbol{\theta}_t}(\mathcal{T}|x)e^{r(x, \mathcal{T})/\tau}}{Z(x, \rho_{\boldsymbol{\theta}_t})}, \quad \text{where} \tag{12}$$

$$Z(x, \rho_{\boldsymbol{\theta}_t}) := \int \rho_{\boldsymbol{\theta}_t}(\mathcal{T}'|x)e^{r(x, \mathcal{T}')/\tau} \, d\mathcal{T}' = \mathbb{E}_{\mathcal{T}' \sim \rho_{\boldsymbol{\theta}_t}(\cdot|x)}[e^{r(x, \mathcal{T}')/\tau}]. \tag{13}$$

This is equivalent to the following: for any pair of trajectories $\mathcal{T}_1$ and $\mathcal{T}_2$,

$$\frac{\rho(\mathcal{T}_1|x)}{\rho(\mathcal{T}_2|x)} = \frac{\rho_{\boldsymbol{\theta}_t}(\mathcal{T}_1|x)}{\pi_{\boldsymbol{\theta}_t}(\mathcal{T}_2|x)} e^{\big(r(x, \mathcal{T}_1) - r(x, \mathcal{T}_2)\big)/\tau}.$$

Taking logarithm of both sides and doing some rearrangement, we have equivalently

$$r(x, \mathcal{T}_1) - \tau \cdot \big( \log \rho(\mathcal{T}_1|x) - \log \rho_{\boldsymbol{\theta}_t}(\mathcal{T}_1|x) \big) = r(x, \mathcal{T}_2) - \tau \cdot \big( \log \rho(\mathcal{T}_2|x) - \log \rho_{\boldsymbol{\theta}_t}(\mathcal{T}_2|x) \big). \tag{14}$$

Note that for a trajectory $\mathcal{T}$, we have

$$\log \rho(\mathcal{T}|x) - \log \rho_{\boldsymbol{\theta}_t}(\mathcal{T}|x) = \sum_\ell \log \pi(a^\ell|c^\ell) - \sum_\ell \log \pi_{\boldsymbol{\theta}_t}(a^\ell|c^\ell)$$

since the state-transition probability terms in $\log \rho(\mathcal{T}|x)$ and $\log \rho_{\boldsymbol{\theta}_t}(\mathcal{T}|x)$ cancel out.

**Step 2: surrogate loss with finite samples.** Given $K$ trajectories from the same initial state $s_1 = x$, we define the following mean-squared surrogate loss that enforces the consistency condition:

$$\widehat{L}(\boldsymbol{\theta}; x, \pi_{\boldsymbol{\theta}_t}) := \frac{1}{K^2} \sum_{1 \le i < j \le K} \frac{(a_i - a_j)^2}{(1 + \tau)^2}, \tag{15}$$

$$\text{where} \quad a_i := r(x, \mathcal{T}_i) - \tau \Big( \sum_\ell \log \pi_{\boldsymbol{\theta}}(a_i^\ell|c_i^\ell) - \sum_\ell \log \pi_{\boldsymbol{\theta}_t}(a_i^\ell|c_i^\ell) \Big). \tag{16}$$

With infinite samples and sufficient coverage of the action space, the optimum of this surrogate loss would be the same as the optimal policy for the surrogate objective in Eq. (11).

**Step 3: one gradient step of the surrogate loss.** By the same trick as in Section 2.2, we have

$$\nabla_{\boldsymbol{\theta}}(a_i - a_j)^2\big|_{\boldsymbol{\theta}_t} = -2\tau \left( r(x, \mathcal{T}_i) - r(x, \mathcal{T}_j) \right) \left( \nabla_{\boldsymbol{\theta}} \sum_{\ell} \log \pi_{\boldsymbol{\theta}}(a_i^\ell | c_i^\ell)\big|_{\boldsymbol{\theta}_t} - \nabla_{\boldsymbol{\theta}} \sum_{\ell} \log \pi_{\boldsymbol{\theta}}(a_j^\ell | c_j^\ell)\big|_{\boldsymbol{\theta}_t} \right),$$

and

$$\nabla_{\boldsymbol{\theta}} \sum_{1 \le i < j \le K} \frac{(a_i - a_j)^2}{(1 + \tau)^2}\bigg|_{\boldsymbol{\theta}_t} = \frac{-2\tau K}{(1 + \tau)^2} \sum_{1 \le i \le K} \left( r(x, \mathcal{T}_i) - \overline{r}(x) \right) \nabla_{\boldsymbol{\theta}} \sum_{\ell} \log \pi_{\boldsymbol{\theta}}(a_i^\ell | c_i^\ell)\big|_{\boldsymbol{\theta}_t},$$

where $\overline{r}(x) \coloneqq \sum_{1 \le j \le K} r(x, \mathcal{T}_j)/K$ denotes the group mean reward in the last line.

In sum, the gradient of the surrogate loss in Eq. (16) becomes:

$$\nabla_{\boldsymbol{\theta}} \widehat{L}(\boldsymbol{\theta}; x, \pi_{\boldsymbol{\theta}_t})\big|_{\boldsymbol{\theta}_t} = \frac{-2\tau}{(1 + \tau)^2} \cdot \frac{1}{K} \sum_{1 \le i \le K} \left( r(x, \mathcal{T}_i) - \overline{r}(x) \right) \nabla_{\boldsymbol{\theta}} \sum_{\ell} \log \pi_{\boldsymbol{\theta}}(a_i^\ell | c_i^\ell)\big|_{\boldsymbol{\theta}_t}.$$

This motivates the following policy update step:

$$g\big(\boldsymbol{\theta}; x, \{\mathcal{T}_i, r_i\}_{1 \le i \le K}\big) = \frac{2\tau}{(1 + \tau)^2} \cdot \frac{1}{K} \sum_{1 \le i \le K} \left( r(x, \mathcal{T}_i) - \overline{r}(x) \right) \nabla_{\boldsymbol{\theta}} \sum_{1 \le \ell \le |\mathcal{T}_i|} \log \pi_{\boldsymbol{\theta}}(a_i^\ell | c_i^\ell), \quad (17)$$

which concludes our derivation of group-relative REINFORCE in multi-step RL settings.

## B  IMPLEMENTATION DETAILS AND ADDITIONAL EXPERIMENTS

We implement all algorithms with the Trinity-RFT framework (Pan et al., 2025), and run experiments on NVIDIA L20, H20, and A800 GPUs. See Tables 1 and 2 for detailed configurations of our experiments.

### B.1  DATASET DETAILS

We provide additional descriptions of the datasets used in our experiments:

- GSM8k (Cobbe et al., 2021) is a widely used benchmark with 8.5k grade-school math word problems, designed to test arithmetic reasoning and step-by-step problem solving.
- MATH (Hendrycks et al., 2021) covers algebra, geometry, probability, and number theory, containing 12.5k examples in total (7.5k for training and 5k for testing); it demands advanced symbolic reasoning beyond GSM8k.
- Guru (Cheng et al., 2025) is a multi-domain reasoning dataset with 91.9k examples spanning math, code, science, logic, simulation, and tabular tasks; we use its math subset (around 54k samples), which introduces diverse problem formats for evaluating transfer of reasoning strategies.
- ToolACE (Liu et al., 2025a) is a multilingual benchmark with around 11k synthetic samples designed to evaluate LLMs' ability to solve tasks by selecting and invoking external tools via strict JSON-formatted function calls; we use a 5k single-turn subset in our experiments.

### B.2  UNDERSTANDING THE SYNCHRONIZATION PARAMETERS

We parameterize rollout-training scheduling by two configuration parameters in Trinity-RFT: the synchronization interval (`sync_interval`) and synchronization offset (`sync_offset`). Their meanings are visualized in Figure 6 and explained in the following.

The parameter `sync_interval` specifies the number of generated rollout batches (which equals the number of gradient steps for training the policy model) between two consecutive executions of model weight synchronization. When `sync_interval = 1`, the rollout and policy models synchronize after each gradient step with one batch of samples, yielding a strictly on-policy process (if we ignore the issue of precision mismatch between rollout and training engines (Yao et al., 2025)). When `sync_interval > 1`, `sync_interval` rollout batches are generated with stale

Table 1: Default hyperparameters. Deviations from defaults are noted in figure captions.

| | GSM8K Qwen2.5 1.5B | ToolACE Llama-3.2 3B | Guru Qwen2.5 7B | Guru Qwen3 30B-A3B | MATH Llama-3.1 8B |
|---|---|---|---|---|---|
| Learning rate | $1\times10^{-6}$ | $1\times10^{-6}$ | $1\times10^{-6}$ | $2\times10^{-6}$ | $5\times10^{-7}$ |
| Batch size | 96 | 96 | 64 | 72 | 64 |
| $K$ | 8 | 8 | 16 | 16 | 16 |
| Weight decay | 0.01 | 0.01 | 0.1 | 0.1 | 0.1 |
| Warmup steps | 0 | 0 | 80 | 80 | 40 |
| Eval temperature | 1.0 | N/A | N/A | N/A | 0.6 |
| Eval top-p | 1.0 | N/A | N/A | N/A | 1.0 |
| Figures | 2, 4, 8, 9, 10 | 3 | 5 | 12 | 11 |

Table 2: Other shared hyperparameters across all experiments.

| Parameter | Value |
|---|---|
| Optimizer | AdamW |
| $(\beta_1, \beta_2)$ | (0.9, 0.999) |
| Gradient clipping | 1.0 |
| Warmup style | constant |
| Weight-decay increment style | constant |
| Auxiliary LR decay style | exponential |
| Training inference temperature | 1.0 |
| Training inference top-p | 1.0 |

model weights before synchronization, which accelerates the overall RL process through pipeline parallelism but incurs off-policyness.

The parameter sync_offset specifies the lag between the generation and consumption of each batch. More specifically, sync_offset batches are generated and saved to the buffer before training is launched, which is also useful for reducing pipeline bubbles and improving hardware utilization (Noukhovitch et al., 2025). In some of our experiments, we deliberately set sync_offset to a large value, in order to simulate a scenario where reward signals from the environment are lagged.

In general, with $(\texttt{sync\_interval}, \texttt{sync\_offset}) = (m, n)$, the off-policyness of a consumed batch with zero-index id $l$ corresponds to its temporal distance from the most recent synchronized policy is $(l \bmod m) + n$. For example, $(4, 0)$ yields off-policyness $0, 1, 2, 3$ within each interval, while $(1, 4)$ yields a constant off-policyness of $4$.

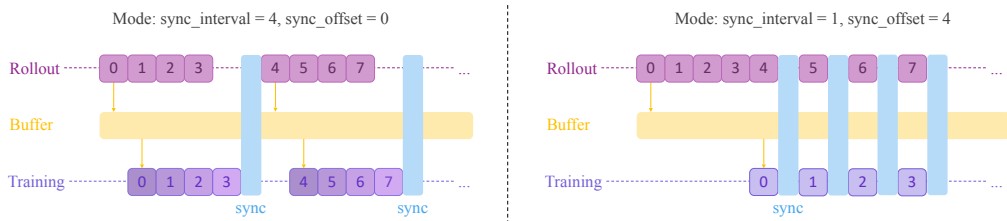

Figure 6: A visualization of the rollout-training scheduling in sync_interval = 4 (left) or sync_offset = 4 (right) modes. Each block denotes one batch of samples for one gradient step, and the number in it denotes the corresponding batch id. Training blocks are color-coded by data freshness, with lighter color indicating increasing off-policyness.

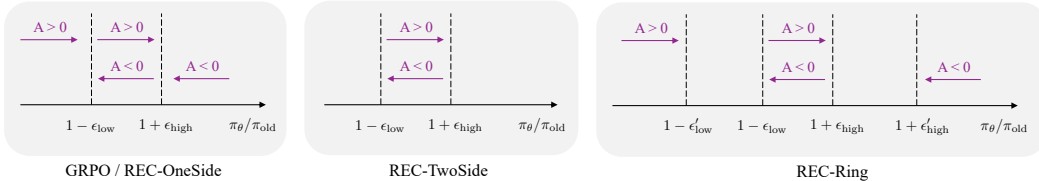

Figure 7: A visualization of activated gradient for various REC algorithms. Here, $A$ represents the advantage of a specific token, and an arrow pointing to the right and annotated with "$A > 0$" means there is activated gradient that incentivizes increasing $\pi_\theta$ when the token advantage is positive and the probability ratio $\pi_\theta/\pi_{\text{old}}$ lies in the corresponding interval.

### B.3 REC WITH DIFFERENT CLIPPING MECHANISMS

In addition to one-side clipping investigated in Section 4, here we compare additional clipping mechanisms for the REC series, to understand how the geometry of clipping — asymmetric vs. symmetric bounds and the presence of a zero-gradient band — affects the learning process.

**REC-TWOSIDE-IS/NOIS.** We replace the mask $M_i^t$ in REC-ONESIDE-IS/NOIS in Eq. (8) with a two-side mask[3]:

$$\widetilde{M}_i^t = \mathbb{1}\left(1 - \epsilon_{\text{low}} \leq \frac{\pi_\theta(y_i^t \mid x, y_i^{<t})}{\pi_{\text{old}}(y_i^t \mid x, y_i^{<t})} \leq 1 + \epsilon_{\text{high}}\right). \tag{18}$$

Two-side clipping imposes weaker regularization than one-side clipping does with the same clipping parameter $(\epsilon_{\text{low}}, \epsilon_{\text{high}})$. This can potentially improve training efficiency, but might also be risky when $\pi_\theta/\pi_{\text{old}}$ goes far off. To compensate for this, we design REC-RING.

**REC-RING.** In addition to the inner band $(1 - \epsilon_{\text{low}}, 1 + \epsilon_{\text{high}})$ as in Eq. (18), we further specify outer safety margins $\epsilon'_{\text{low}} \geq \epsilon_{\text{low}}$ and $\epsilon'_{\text{high}} \geq \epsilon_{\text{high}}$. The REC-RING mask is:

$$\widehat{M}_i^t = \mathbb{1}\left(1 - \epsilon_{\text{low}} \leq \frac{\pi_\theta(y_i^t \mid x, y_i^{<t})}{\pi_{\text{old}}(y_i^t \mid x, y_i^{<t})} \leq 1 + \epsilon_{\text{high}}\right) \tag{19}$$

$$+ \mathbb{1}\left(A_i > 0 \text{ and } \frac{\pi_\theta(y_i^t \mid x, y_i^{<t})}{\pi_{\text{old}}(y_i^t \mid x, y_i^{<t})} \leq 1 - \epsilon'_{\text{low}}\right) \tag{20}$$

$$+ \mathbb{1}\left(A_i < 0 \text{ and } \frac{\pi_\theta(y_i^t \mid x, y_i^{<t})}{\pi_{\text{old}}(y_i^t \mid x, y_i^{<t})} \geq 1 + \epsilon'_{\text{high}}\right). \tag{21}$$

A comparison of the clipping mechanisms are visualized in Figure 7. Note that REC-ONESIDE and REC-TWOSIDE can be regarded as special cases of REC-RING.

**Experiments.** We compare the following algorithms: REINFORCE, GRPO, REC-TWOSIDE-IS, REC-TWOSIDE-NOIS, and REC-RING-NOIS. Clipping parameters are set to $(\epsilon_{\text{low}}, \epsilon_{\text{high}}) = (0.2, 0.2)$, and for REC-RING we additionally set $(\epsilon'_{\text{low}}, \epsilon'_{\text{high}}) = (0.6, 2.0)$.

Figure 8 presents the empirical results. We observe that for REC-TWOSIDE, importance sampling is non-essential in all three settings, akin to the case of REC-ONESIDE. In addition, REC-TWOSIDE methods demonstrate fast policy improvement at the beginning but tend to collapse later on, whereas REC-RING achieves a better balance of convergence speed and stability.

### B.4 ABLATION: THE IMPACT OF LEARNING RATES

Recall that in Section 4.1, we have demonstrated empirically the advantages of enlarging the clipping parameters $\epsilon_{\text{low}}, \epsilon_{\text{high}}$ for REC-ONESIDE-NOIS. One might wonder if the relatively weak

---

[3]It turns out that REC-TWOSIDE-NOIS resembles the sPPO algorithm proposed by Vaswani et al. (2022), though derived with different rationales.

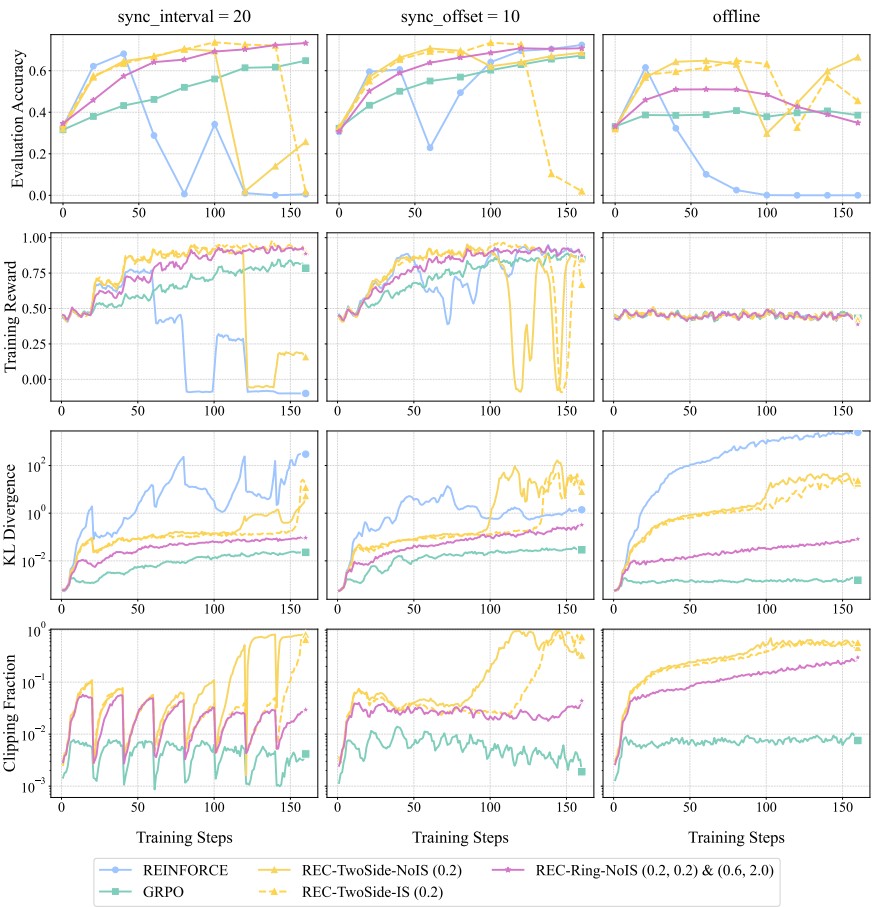

Figure 8: Comparison of REC variants on GSM8K with Qwen2.5-1.5B-Instruct under different off-policy settings. Evaluation accuracy, training reward, KL divergence (with respect to the initial model) and clipping fraction are reported. Training reward curves are smoothed with a running-average window of size 3.

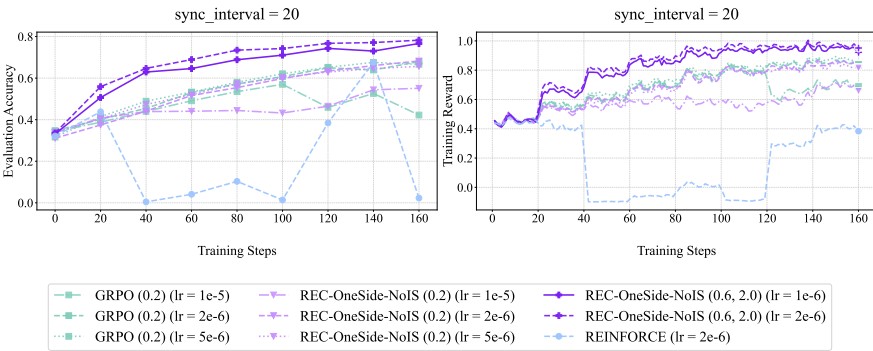

Figure 9: Comparison of GRPO and REC-ONESIDE-NOIS on GSM8K with Qwen2.5-1.5B-Instruct. Evaluation accuracy (left) and training reward (right) are reported for varying learning rates.

performance of GRPO or REC-ONESIDE with conventional $\epsilon_{\text{low}} = \epsilon_{\text{high}} = 0.2$ is genuinely rooted in the clipping mechanism itself, or simply due to the choice of a small learning rate.

To answer this, we enhance the experiment of Figure 2 by sweeping learning rates over $\{1\times10^{-5}, 2\times 10^{-6}, 5\times10^{-6}\}$. The results are illustrated in Figure 9, which confirm that simply increasing the learning rate cannot bridge the performance gap between GRPO with $\epsilon_{\text{low}} = \epsilon_{\text{high}} = 0.2$ and REC-ONESIDE-NOIS with $\epsilon_{\text{low}} = 0.6, \epsilon_{\text{high}} = 2.0$. This shows that relaxing the clipping range acts as a genuine improvement of regularization, rather than merely mimicking a larger learning rate.

*Remark* 1. Let us elaborate on the initial motivation and rationale behind our investigation of enlarging the clipping range. In the conventional theory of importance sampling and bias-variance trade-off for policy gradient estimation, we need certain trust-region condition that constrains the distance between the actor policy $\pi_{\theta}$ and behavior policy $\pi_{\text{old}}$. By this rationale, LLM-RL frameworks commonly adopt small values $\epsilon_{\text{low}} = \epsilon_{\text{high}} = 0.2$ for the default clipping range, or relax it up to $\epsilon_{\text{high}} = 0.28$ as proposed by DAPO (Yu et al., 2025). Our work, in contrast, proposes a native off-policy interpretation for group-relative REINFORCE. The essential rationale is no longer about unbiased policy gradient estimation, and importance sampling is found to be non-essential. While clipping remains a valid option (among many others) for regularization, we are curious about whether the conventional choice of narrow clipping range is still necessary, especially after observing that GRPO's learning progress slows down drastically (as the fraction of clipped tokens grows) with increasing off-policyness. Conceptually, our off-policy interpretation would advocate a larger clipping range than allowed by the conventional policy gradient theory. We hope that our preliminary exploration of this aspect could inspire the community to rethink the true working mechanism behind GRPO-like algorithms and the possibility of faster off-policy learning.

## B.5 EXPERIMENTS FOR OPMD AND ASYMRE

Figure 10 presents empirical results for OPMD and AsymRE in various off-policy settings. It is worth noting that, while the analysis and experiments in their original papers (Kimi-Team, 2025b; Arnal et al., 2025) focus on a setting that is effectively the same as our `sync_interval > 1` setting, our analysis and experiments have also validated their efficacy in `sync_offset > 0` scenarios.

## B.6 ADDITIONAL DETAILS AND RESULTS FOR RED ALGORITHMS

We present further implementation details for the RED-DROP and RED-WEIGHT algorithms investigated in Section 4.3:

- RED-DROP: When the number of negative samples in a group exceeds the number of positive ones, we randomly drop the excess negatives so that positives and negatives are balanced. After this subsampling step, we recompute the advantages using the remaining samples, which are then fed into the loss.

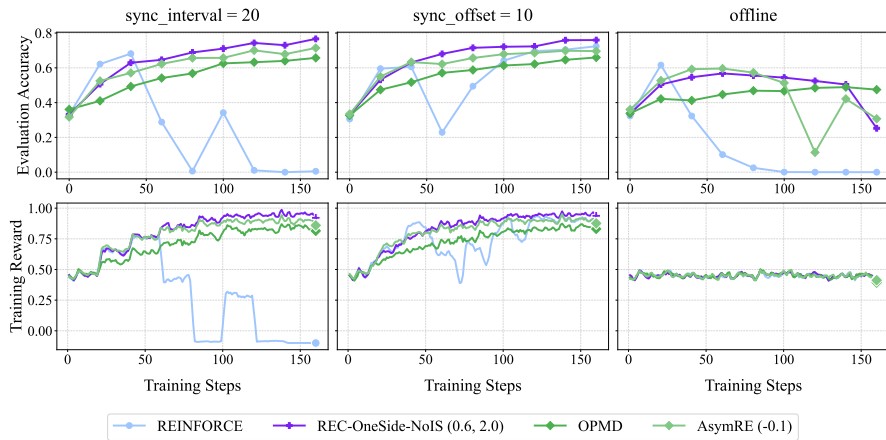

Figure 10: Empirical results for OPMD and AsymRE (cf. Section 4.2) on GSM8K with Qwen2.5-1.5B-Instruct under various off-policy settings. The regularization coefficient for OPMD and the baseline shift for AsymRE are both 0.1. Training reward curves are smoothed with a running-average window of size 3.

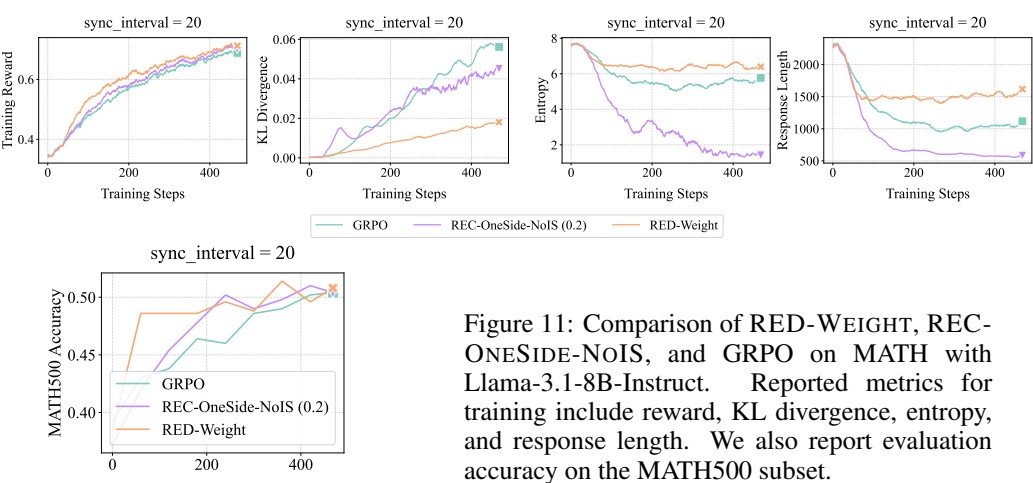

Figure 11: Comparison of RED-WEIGHT, REC-ONESIDE-NOIS, and GRPO on MATH with Llama-3.1-8B-Instruct. Reported metrics for training include reward, KL divergence, entropy, and response length. We also report evaluation accuracy on the MATH500 subset.

- RED-WEIGHT: Each sample $i$ is weighted by $w_i = \exp(A_i/T)$, where $A_i$ denotes its advantage estimate and $T > 0$ is a temperature parameter controlling the sharpness of weighting. Intuitively, this scheme amplifies high-advantage samples while down-weighting low-advantage ones. We fix $T = 1$ for all experiments.

Additional experiments for RED-WEIGHT, and its comparison against GRPO and REC-ONESIDE-NOIS, can be found in Figure 11. We observe that for the MATH dataset and Llama-3.1-8B-Instruct, RED-WEIGHT achieves higher rewards with lower KL divergence, while maintaining more stable entropy and response lengths.

## B.7 GSPO: SEQUENCE-LEVEL IMPORTANCE SAMPLING COULD BE NON-ESSENTIAL

Group Sequence Policy Optimization (GSPO) (Zheng et al., 2025) proposes to replace token-wise clipping and importance sampling in GRPO with sequence-wise counterparts. Similar to Finding F1 in Section 4 for GRPO, we hypothesize that GSPO's effectiveness stems from sequence-level clipping as regularization, rather than from sequence-level importance sampling. We provides preliminary validation for this hypothesis, through experiments with GSPO-style REC variants.

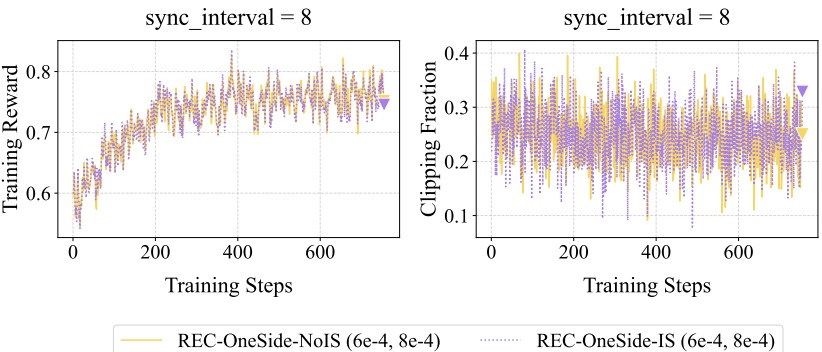

Figure 12: Empirical results on Guru-Math with Qwen3-30B-A3B (MoE). Training reward curves are smoothed with a running-average window of size 3.

**Implementations.** Given a prompt $x$ and $K$ responses $\{y_i\}_{1 \le i \le K}$, let $s_i(\boldsymbol{\theta})$ denote the length-normalized sequence-level probability ratio for $y_i$:

$$s_i(\boldsymbol{\theta}) := \left( \frac{\pi_{\boldsymbol{\theta}}(y_i \mid x)}{\pi_{\mathsf{old}}(y_i \mid x)} \right)^{\frac{1}{|y_i|}} = \exp\left( \frac{1}{|y_i|} \sum_{1 \le t \le |y_i|} \log \frac{\pi_{\boldsymbol{\theta}}(y_i^t \mid x, y_i^{<t})}{\pi_{\mathsf{old}}(y_i^t \mid x, y_i^{<t})} \right).$$

We further define the one-side sequence-level clipping mask as

$$M_i := \mathbb{1}\left( A_i > 0, \ s_i(\boldsymbol{\theta}) \le 1 + \epsilon_{\mathsf{high}} \right) + \mathbb{1}\left( A_i < 0, \ s_i(\boldsymbol{\theta}) \ge 1 - \epsilon_{\mathsf{low}} \right).$$

With these notations in place, we implement two GSPO-style REC variants as follows:

$$\text{REC-GSPO-IS: } \boldsymbol{g} = \frac{1}{K} \sum_{1 \le i \le K} \frac{1}{|y_i|} \sum_{1 \le t \le |y_i|} \nabla_{\boldsymbol{\theta}} \log \pi_{\boldsymbol{\theta}}(y_i^t \mid x, y_i^{<t}) \cdot (r_i - \overline{r}) \, s_i(\boldsymbol{\theta}) \, M_i,$$

$$\text{REC-GSPO-NoIS: } \boldsymbol{g} = \frac{1}{K} \sum_{1 \le i \le K} \frac{1}{|y_i|} \sum_{1 \le t \le |y_i|} \nabla_{\boldsymbol{\theta}} \log \pi_{\boldsymbol{\theta}}(y_i^t \mid x, y_i^{<t}) \cdot (r_i - \overline{r}) \, M_i.$$

One can check that REC-GSPO-IS is equivalent to GSPO (except that we use $r_i - \overline{r}$ as the advantage, without normalization by $\sigma_r$), while REC-GSPO-NoIS discards the sequence-level importance-sampling weights.

**Experiments.** We use the Guru-Math dataset and a mixture-of-expert (MoE) model — Qwen3-30B-A3B (Qwen-Team, 2025b) — since stable RL for MoE models is one of the main motivations behind GSPO (Zheng et al., 2025). We set `sync_interval` = 8, $\epsilon_{\mathsf{low}} = 6 \times 10^{-4}$, and $\epsilon_{\mathsf{high}} = 8 \times 10^{-4}$; other hyperparameters can be found in Tables 1 and 2.

Figure 12 shows that the learning curves of both REC-GSPO variants — with or without importance sampling — mostly overlap, indicating that importance sampling is likely a non-essential component for the effectiveness of GSPO.

## C   SUMMARY: A UNIFIED VIEW OF VARIOUS ALGORITHMS

For convenient reference, Table 3 summarizes the algorithms investigated in Section 4.

Table 3: A summary of algorithms investigated in Section 4.

| Augmentation | Algorithm | Gradient / Loss |
|---|---|---|
| Regularize by clipping | GRPO | $g = \frac{1}{K}\sum_i \sum_t \nabla_\theta \log \pi_\theta(y_i^t \mid x, y_i^{<t}) \cdot A_i \frac{\pi_\theta(y_i^t \mid x, y_i^{<t})}{\pi_{\text{old}}(y_i^t \mid x, y_i^{<t})} M_i^t$ |
| | REC-ONESIDE-IS | $g = \frac{1}{K}\sum_i \sum_t \nabla_\theta \log \pi_\theta(y_i^t \mid x, y_i^{<t}) \cdot (r_i - \bar{r}) \cdot \frac{\pi_\theta(y_i^t \mid x, y_i^{<t})}{\pi_{\text{old}}(y_i^t \mid x, y_i^{<t})} M_i^t$ |
| | REC-ONESIDE-NOIS | $g = \frac{1}{K}\sum_i \sum_t \nabla_\theta \log \pi_\theta(y_i^t \mid x, y_i^{<t}) \cdot (r_i - \bar{r}) M_i^t$ |
| Add regularization loss | OPMD | $\widehat{L} = -\frac{1}{K}\sum_i (r_i - \bar{r})\log \pi_\theta(y_i \mid x) + \frac{\beta}{2K}\sum_i (\log \pi_\theta(y_i \mid x) - \log \pi_{\text{old}}(y_i \mid x))^2$ |
| | AsymRE | $\widehat{L} = -\frac{1}{K}\sum_i (r_i - \bar{r})\log \pi_\theta(y_i \mid x) - \frac{\beta}{K}\sum_i \log \pi_\theta(y_i \mid x)$ |
| Reweight data | Pairwise-weighted REINFORCE | $g = \frac{1}{K}\sum_i \left(\sum_j w_{i,j}\right)\left(r_i - \frac{\sum_j w_{i,j} r_j}{\sum_j w_{i,j}}\right)\nabla_\theta \log \pi_\theta(y_i \mid x)$ |
| | RED-Drop | $g = \frac{1}{|S|}\sum_{i \in S}(r_i - \bar{r}_S)\nabla_\theta \log \pi_\theta(y_i \mid x)$ |
| | RED-Weight | $g = \sum_i w_i(r_i - \bar{r})\nabla_\theta \log \pi_\theta(y_i \mid x), \quad w_i = \exp(A_i/T)$ |

## D  ADDITIONAL RELATED WORKS

We focus our discussion on prior works that are most closely related to our core analysis in Section 2.2. In a tabular setting, the surrogate objective in Eq. (3) — KL-regularized reward maximization — can be regarded as an instantiation of mirror descent, whose optimum admits the closed form in Eq. (4). In more general settings with parameterized policy $\pi_\theta$ and large action space, it is infeasible to realize Eq. (4) directly, and one would resort to optimizing the model parameters. Various algorithms have been developed on the basis of Eq. (3) and (4), including Kimi's OPMD (Kimi-Team, 2025b) as explained in Section 4.2.

REBEL (Gao et al., 2024) has a derivation that largely overlaps with our Step 1 and 2 analysis in Section 2.2. It then seeks to solve the squared loss in Eq. (6), which enforces the pairwise consistency condition in Eq. (5). CoPG (Flet-Berliac et al., 2024) takes a similar approach, except that it uses a fixed reference policy (rather than the current iteration $\pi_{\theta_t}$) for KL regularization. Compared to REINFORCE-style algorithms — for which enterprise-grade LLM-RL frameworks like verl (Sheng et al., 2024) and Trinity-RFT (Pan et al., 2025) have been heavily optimized for — REBEL and CoPG could be less infrastructure-friendly or efficient. For example, in the presence of data parallelism and gradient accumulation, these frameworks can automatically divide a mini-batch into multiple micro-batches (each containing multiple or just one sequence) in a way that maximizes load balancing and training efficiency, while minimizing peak memory usage. However, solving the squared loss in Eq. (6) (like REBEL does) contradicts these performance optimization techniques, as it requires paired responses for the same prompt to be located within the same micro-batch. This constraint increases infrastructure complexity and peak memory usage, as reported by Brantley et al. (2025). Our Step 3 analysis in Section 2.2, on the other hand, proposes to take one gradient descent step for the squared loss, leading to a group-relative variant of classic REINFORCE while giving it a native off-policy interpretation.

Natural Policy Gradient (NPG) (Kakade, 2001) can be derived by approximating the surrogate objective in Eq. (3) with first-order Taylor expansion for the max-reward term and second-order Taylor expansion for the KL term, and then setting its gradient to zero. Since NPG requires on-policy sampling, it is less relevant to our study of off-policy LLM-RL. DPO (Rafailov et al., 2023) was also derived on the basis of Eq. (3), (4) and (5), but in a substantially different setting, with pairwise preference data and the Bradley-Terry assumption.

