# OpenReview forum: "Group-Relative REINFORCE Is Secretly an Off-Policy Algorithm: Demystifying Some Myths About GRPO and Its Friends"
_ICLR.cc/2026/Conference — ICLR 2026 Poster_

### Official Review · Reviewer_5aHq · 2025-10-29

**Soundness:** 3
**Presentation:** 3
**Contribution:** 3
**Rating:** 6
**Confidence:** 3

**Summary:**

This paper presents a new perspective that REINFORCE-based algorithms, including GRPO, can be interpreted as inherently 'off-policy' algorithms with a solid theoretical foundation. Based on this off-policy interpretation, it proposes two core principles for stable learning: regularizing policy updates and actively shaping the data distribution. It frames previously disparate GRPO-like RL algorithms into a single, unified framework, proposing several new findings and using this interpretation to suggest a insight about how performance can be improved.

**Strengths:**

1. The paper's core contribution is its rigorous 'first-principles derivation' that recasts GRPO as a native off-policy algorithm. This reinterprets GRPO(and its friends) not as a on-policy method but as a principled 1-step gradient descent on a KL-regularized surrogate objective.

2. The paper successfully reinterprets and unifies recent algorithms like Kimi's OPMD and Meta's AsymRE. This demonstrates the framework's utility as an analytical tool beyond just GRPO, offering a new lens through which to understand a broader class of REINFORCE-based methods.

3. The paper provides a solid theoretical justification for previously heuristic data shaping techniques, such as "dropping negatives" (RED-DROP) or "up-weighting positives" (RED-WEIGHT). By showing that these established methods can be formally derived from their proposed loss, the authors successfully frame them as principled components of their unified theory, rather than just ad-hoc empirical tricks.

**Weaknesses:**

1. As the authors acknowledge, the paper proposes a novel interpretation but does not provide a theoretical bridge guaranteeing convergence to an optimal policy.


2. The proposal to widen the clipping range (e.g., to (0.6, 2.0)) is presented purely as an empirical finding without theoretical backing (unlike the data shaping principle). The paper provides no detailed analysis on why this wider range is effective, whether it relates to task properties, or how one might determine an optimal "loose" range etc. Even though, Appendix B.4 shows that widening the clipping range is different from just controlling learning rate, it still remains a task-specific hyperparameter that requires empirical tuning, rather than a generalizable principle.

**Questions:**

1. You claim that token-level IS is "non-essential," but your appendix figures demonstrate different aspects between NoIS and IS. Why do you think these phenomena have been observed?

2. Your paper argues that at the token level, clipping is important for stable training of GRPO[1] while IS is "non-essential." Given recent work like GSPO, which moves to a sequence-level framework, do you hypothesize that your findings(especially finding1) generalize? Would you predict that sequence-level IS can also be interpreted as non-essential in that framework under off-policy settings?

[1] Chujie Zheng, Shixuan Liu, Mingze Li, Xiong-Hui Chen, Bowen Yu, Chang Gao, Kai Dang, Yuqiong Liu, Rui Men, An Yang, Jingren Zhou, & Junyang Lin. (2025). Group Sequence Policy Optimization.

---

> ### Author Response · Authors · 2025-11-22
> **Response to Reviewer 5aHq (1/2)**
>
> Thank you for your thoughtful and constructive review! We address your questions and concerns in the following.
>
> ---
>
> **Question: Does Finding 1 generalize to sequence-level clipping and importance sampling, like in GSPO?**
>
> Thank you for this interesting question!
> Our short answer is yes: Finding 1 can generalize to GSPO, and in particular, sequence-level IS in GSPO could be non-essential (i.e., not absolutely necessary) as well.
> This is elaborated in the newly added Appendix B.7 of our revised manuscript, and also in the following.
>
> To answer your question, we have implemented and tested two REC variants --- REC-GSPO-IS and REC-GSPO-NoIS --- during the rebuttal period.
> These variants are similar to REC-OneSide-IS/NoIS presented in Section 4.1, with three major differences:
> + Clipping is done at the sequence level: the token-wise probability ratio in the one-side clipping mask defined in Eq. (8) of our manuscript is replaced by the length-normalized sequence-wise probability ratio $s_i(\theta)$ defined in Eq. (7) of the GSPO paper.
> + For REC-GSPO-IS, the IS weights are set to $s_i(\theta)$, whereas REC-GSPO-NoIS discards these weights.
> + Following the original GSPO paper, REC-GSPO-IS/NoIS adopt "seq-mean-token-mean" loss aggregation (i.e., loss terms are first averaged across tokens within each sequence, then averaged across sequences), in contrast to "token-mean" aggregation adopted by REC-OneSide-IS/NoIS (cf. Eq. (1b) of our manuscript).
>
> One can check that REC-GSPO-IS is equivalent to GSPO, just like REC-OneSide-IS being equivalent to GRPO (except that we don't normalize advantages by the standard deviation of rewards).
>
> We conduct new experiments for these new REC variants on Guru-Math with Qwen3-30B-A3B (a mixture-of-expert model, since stable RL for MoE models is one of the main motivations behind GSPO).
> Empirical results have been added to Figure 12 in our revised manuscript.
> We observe that REC-GSPO-NoIS works perfectly well (and its reward curve largely overlaps with that of REC-GSPO-IS), which confirms that the sequence-level IS weights in GSPO indeed can be safely discarded in this case.
> This supports our hypothesis that sequence-level clipping is another effective way of regularization for augmenting group-relative REINFORCE (in line with the regularization principle presented in Section 3), whereas sequence-level IS might play a non-essential role in GSPO.
>
> ---
>
> **Question: "You claim that token-level IS is 'non-essential,' but your appendix figures demonstrate different aspects between NoIS and IS. Why do you think these phenomena have been observed?"**
>
> Thank you for this question.
>
> + To clarify, throughout this work, "non-essential" literally means "not absolutely necessary" (rather than "making no impact"). That is, by claiming "token-level IS in GRPO is non-essential", we mean to say that importance sampling is not absolutely necessary for effective off-policy learning, which has been backed by our theoretical analysis and validated by the empirical success of REC-OneSide-NoIS across diverse settings.
> Therefore, deviation among the learning curves of different algorithms (with or without IS) doesn't necessarily contradict our claim.
>
> + In addition to the effects of IS weights, another source of deviation between REC-OneSide-IS versus NoIS is the inherent randomness of different experiment trials.
> Regarding REC-OneSide-NoIS versus GRPO, another source of deviation is that GRPO normalizes advantages by the standard deviation of rewards.
>
> + Figure 8 in the appendix presents the learning curves of REC-*TwoSide*-IS versus NoIS in the same plots. We want to clarify that Finding F1 (including the claim about token-level IS being non-essential) is tailored to GRPO and REC-OneSide-IS/NoIS, thus not necessarily applicable to the REC-TwoSide variants.

---

> ### Author Response · Authors · 2025-11-22
> **Response to Reviewer 5aHq (2/2)**
>
> **Weakness: lack of detailed analysis for enlarging the clipping range.**
>
> We acknowledge that in the current work, the effectiveness of enlarging the clipping range appears more like an empirical finding with limited theoretical backing.
> Let us further elaborate on the initial motivation and rationale behind our investigation of this approach:
>
> + In the conventional theory of importance sampling and bias-variance trade-off for policy gradient estimation, we need certain trust-region condition that constrains the distance between the actor policy $\pi_{\theta}$ and behavior policy $\pi_{\text{old}}$.
> By this rationale, LLM-RL frameworks commonly adopt small values $\epsilon_{\text{low}} = \epsilon_{\text{high}} = 0.2$ for the default clipping range, or relax it up to $\epsilon_{\text{high}} = 0.28$ as proposed by DAPO [1].
>
> + Our work, in contrast, proposes a *native off-policy interpretation* for group-relative REINFORCE.
> The essential rationale is no longer about unbiased policy gradient estimation, and importance sampling is found to be non-essential.
> While clipping remains a valid option (among many others) for regularization, we are curious about whether the conventional choice of narrow clipping range is still necessary, especially after observing that GRPO's learning progress slows down drastically (as the fraction of clipped tokens grows) with increasing off-policyness.
> Conceptually, our off-policy interpretation would advocate a larger clipping range than allowed by the conventional policy gradient theory.
>
> Motivated by these, we conducted experiments for REC-OneSide-IS/NoIS with aggressive clipping range $\epsilon_{\text{low}} = 0.6$ and $\epsilon_{\text{high}} = 2$ (one magnitude larger than the conventional choice!), and were surprised to find that it actually works well in many cases, with fast and stable convergence.
>
> While our exploration of this aspect is still preliminary, we believe that such a finding could be valuable and deserve to be shared with the community, as it might inspire people to rethink the true working mechanism behind GRPO-like algorithms and the possibility of faster off-policy learning.
> If the reviewer finds the above explanations useful, we will incorporate them into our revision.
> Designing methods for automatically finding the optimal clipping range in a particular setting, on the other hand, is beyond the scope of our current focus in this work.
>
> [1] DAPO: An Open-Source LLM Reinforcement Learning System at Scale. arXiv 2025.
>
>
> ---
>
> **Weakness: "As the authors acknowledge, the paper proposes a novel interpretation but does not provide a theoretical bridge guaranteeing convergence to an optimal policy."**
>
> Indeed, our focus in the current work is to present the most essential rationale for the proposed off-policy interpretation, its numerous algorithmic implications, and extensive empirical evidence that supports our analytical findings.
> Proving formal convergence guarantees --- while identifying weak and realistic assumptions about off-policy data distribution --- is another story.
> We anticipate that it could be fairly challenging, and might need to build upon a theoretical foundation that is substantially different from those (e.g., policy gradient theory) in prior literature.
> We thus refrain from attempting to give partial theoretical results, in order to avoid the risk of misleading others.
> Instead, we acknowledge that rigorous theoretical guarantees for convergence to the optimal policy are beyond the scope of our current focus, and hope that our work would inspire further exploration and research along this promising direction.
>
> ---
>
> Thank you again for your review, which helps us a lot in improving the current work! We hope that our responses have provided satisfactory answers to your questions, but do let us know if you have any further concern that we can address.

---

### Official Review · Reviewer_ZsF7 · 2025-11-01

**Soundness:** 3
**Presentation:** 3
**Contribution:** 2
**Rating:** 4
**Confidence:** 3

**Summary:**

This paper presents a novel theoretical interpretation of Group-Relative REINFORCE (GRPO), showing that it naturally admits an off-policy formulation without assuming on-policy data. The authors derive this view from first principles, unify several recent algorithms (such as Kimi’s OPMD and Meta’s AsymRE) under the same framework, and support the theory with extensive experiments on reasoning and tool-use benchmarks.

**Strengths:**

- Provides a fresh and insightful theoretical reinterpretation of GRPO as an off-policy algorithm.
- Unifies several recent algorithms (e.g., OPMD, AsymRE) under a consistent framework.
- Writing is clear.

**Weaknesses:**

- Innovation lies mainly in conceptual reinterpretation, rather than proposing new algorithms.
- The relation to DPO, Mirror Descent, and other gradient-based RL theories could be expanded.

**Questions:**

- I think the writing style of this paper is quite different from that of typical research papers. Could the authors provide a more systematic discussion of prior work built upon this foundation and establish clearer connections to previous methods?
- The paper mainly analyzes existing group-based algorithms from a mechanistic perspective. If the authors are unable to design more effective and efficient algorithms based on their proposed theory, then the contribution of this work appears to be overstated.

---

> ### Author Response · Authors · 2025-11-22
> **Response to Reviewer ZsF7 (1/2)**
>
> Thank you for your thoughtful and constructive review! We address your questions and concerns in the following.
>
> ---
>
> **Suggestion: provide more systematic discussion of prior work, establish clearer connections to previous methods.**
>
> Thank you for the suggestion!
>
> In our revised manuscript, we have expanded "related works" into a full section.
> The first half of it provides a high-level overview of diverse perspectives for off-policy LLM-RL, similar to the "related works" paragraph in our original submission.
> **The second half is new**: it provides a systematic and detailed discussion about previous methods that are most closely related to our work, in particular, our analysis for off-policy RL in Section 2.2:
>
> + In a tabular setting, the surrogate objective in Eq. (3) --- KL-regularized reward maximization --- can be regarded as an instantiation of mirror descent,
> whose optimum admits the closed-form solution in Eq. (4).
>
> + In more general settings where the policy $\pi$ is parameterized by $\theta$ and the action space is large, it is infeasible to realize the optimal policy directly, and one would resort to optimizing the model parameters $\theta$. Various algorithms have been developed on the basis of Eq. (3) and (4):
>
>   + Kimi's OPMD [1], as explained in Section 4.2, was initially derived by taking logarithm of both sides of Eq. (4) and constructing a squared loss that enforces the equality, followed by two approximations that simplify the formula.
>
>   + REBEL's derivation [2] largely overlaps with our analysis in Steps 1 and 2 of Section 2.2. It then seeks to solve the squared loss in Eq. (6), which enforces the pairwise consistency condition in Eq. (5).
>   CoPG [3] takes a similar approach, except that it uses a fixed reference policy (rather than the current iteration $\pi_{\theta_{t}}$) for KL regularization.
>   Compared to REINFORCE-style algorithms, these methods add to infrastructure complexity and could be less computationally efficient, as explained in our revised Section 5.
>
>   + Our Step 3 analysis, on the other hand, proposes to take one gradient descent step for the squared loss in Eq. (6).
>   This leads to a group-relative variant of classic REINFORCE, while giving it a native off-policy interpretation.
>
>   + Natural Policy Gradient (NPG) [4] can be derived by approximating the surrogate objective in Eq. (3) with first-order Taylor expansion for the reward-maximization term and second-order Taylor expansion for the KL term, and then setting its gradient to zero. Since NPG requires on-policy sampling, it is less relevant to our study of off-policy LLM-RL.
>
>   + DPO [5] was originally derived on the basis of Eq. (3), (4) and (5) as well. It is also less relevant to our work, since it considers a substantially different setting, with pairwise preference data and the Bradley-Terry-model assumption.
>
> We hope that these could help clarify the connections between our work and the literature.
> Please feel free to let us know if you have further comments or suggestions for our revised "related works" section.
>
>
> [1] Kimi k1.5: Scaling reinforcement learning with LLMs. arXiv, 2025.
>
> [2] REBEL: Reinforcement Learning via Regressing Relative Rewards. NeurIPS, 2024.
>
> [3] Contrastive Policy Gradient: Aligning LLMs on sequence-level scores in a supervised-friendly fashion. EMNLP, 2024.
>
> [4] A Natural Policy Gradient. NeurIPS, 2001.
>
> [5] Direct preference optimization: Your language model is secretly a reward model. NeurIPS, 2023.

---

> ### Author Response · Authors · 2025-11-22
> **Response to Reviewer ZsF7 (2/2)**
>
> **Weakness and question: "Innovation lies mainly in conceptual reinterpretation, rather than proposing new algorithms." "If the authors are unable to design more effective and efficient algorithms based on their proposed theory, then the contribution of this work appears to be overstated."**
>
>
> Although there are some new algorithmic ingredients (such as removing the importance-sampling weights or enlarging the clipping range of GRPO) in this work,
> our main focus and innovation are indeed more about novel analysis and reinterpretation of existing algorithms, rather than proposing new ones.
> We hope that our work could make timely contributions to the community, by bringing new insights and clarity to our understanding of foundational and widely adopted algorithms like GRPO (and other variants of group-relative REINFORCE), which could potentially inspire further developments of more effective and efficient algorithms for off-policy LLM-RL.
>
> Regarding the reviewer's question: it is definitely not our intention to overstate the contribution of this work.
> It is greatly appreciated if the reviewer could elaborate on which parts of our manuscript could give readers the wrong impression about our claimed contribution;
> we strive to revise our manuscript in order to avoid such confusion.
> For example, in the last sentence of the introduction section, we may rephrase "our work opens up new opportunities for [...] algorithm design in [...]" as "we hope that our work will inspire further research on [...] algorithm design in [...]", if the reviewer finds this useful.
>
> ---
>
> Thank you again for your review, which helps us a lot in improving the current work! We hope that our responses have provided satisfactory answers to your questions, but do let us know if you have any further concern that we can address.

---

> > ### Comment · Reviewer_ZsF7 · 2025-11-25
> >
> > Thank you for your response. The authors acknowledge that the main innovation of this paper lies in a conceptual reinterpretation rather than in proposing new algorithms. Therefore, I believe the contribution is below the acceptance threshold unless the authors can develop more effective algorithms based on their theory. I will thus maintain my score and encourage the authors to participate in a new round of review.

---

> > > ### Author Response · Authors · 2025-11-27
> > > **Response to Reviewer ZsF7**
> > >
> > > Thank you for your reply and for sharing your perspective.
> > >
> > > **Concern: "I believe the contribution is below the acceptance threshold unless the authors can develop more effective algorithms based on their theory."**
> > >
> > > We respectfully disagree.
> > > **As far as we understand, ICLR is an inclusive community that welcomes research papers of diverse types**, including papers that focus on developing novel algorithms and achieving SOTA scores, as well as papers that focus on advancing understanding, providing theoretical insights, or reinterpreting existing methods in ways that facilitate future progress.
> > > This can be seen from the official [ICLR Reviewer Guide](https://iclr.cc/Conferences/2026/ReviewerGuide), for example:
> > >
> > > + (Regarding significance and contribution) *"Note, this does not necessarily require state-of-the-art results. Submissions bring value to the ICLR community when they convincingly demonstrate new, relevant, impactful knowledge (incl., empirical, theoretical, for practitioners, etc)."*
> > > + *"We encourage reviewers to be broad in their definitions of originality and significance. For example, originality may arise from [...] removing limitations from prior results."* (Indeed, one of our contributions is to remove the conventional on-policy assumption for group-relative REINFORCE and its variants.)
> > > + *"Be mindful of potential biases and try to be open-minded about the value and interest a paper can hold for the entire ICLR community, even if it may not be very interesting for you."*
> > >
> > > As acknowledged in your initial review, our work provides a *"fresh and insightful reinterpretation"* for GRPO and several other group-based RL algorithms in off-policy settings, under a unified and consistent framework.
> > > We believe that deepening the understanding of the mechanisms behind widely used algorithms is no less important than developing new ones.
> > > Moreover, the former could bring helpful insights to the latter, for example:
> > >
> > > + Our theory justifies removing the importance-sampling weights in GRPO, as validated empirically with our proposed REC-OneSide-NoIS algorithm. This could potentially improve the algorithm's robustness to off-policyness caused by the precision mismatch between rollout and training engines [1], although a comprehensive study is beyond our current focus and left for future work.
> > > + Our analysis in Section 4.3 offers principled justification for actively reshaping the data distribution. This could open opportunities for exploring more advanced techniques of data filtering or re-sampling (like those in [2, 3] and beyond), without worry about breaking the assumptions of conventional policy gradient theory.
> > >
> > > It is indeed intriguing future work to design more effective algorithms based on our proposed theory,
> > > and we hope that our work will serve as a good starting point.
> > >
> > >
> > > [1] https://fengyao.notion.site/off-policy-rl
> > >
> > > [2] https://moonshotai.github.io/Kimi-Researcher
> > >
> > > [3] https://arxiv.org/abs/2508.20722

---

### Official Review · Reviewer_5sDW · 2025-11-03

**Soundness:** 3
**Presentation:** 2
**Contribution:** 3
**Rating:** 4
**Confidence:** 4

**Summary:**

This paper argues that popular reinforcement learning (RL) algorithms for LLMs, like Group Relative Policy Optimization (GRPO), are mistakenly treated as on-policy methods. The authors provide a new theoretical derivation showing that group-relative REINFORCE (the basis for GRPO) is "secretly an off-policy algorithm".

**Strengths:**

The paper's main contribution is a new theoretical framework that re-interprets group-relative REINFORCE algorithms, such as GRPO, as natively off-policy. While these methods are typically viewed as on-policy algorithms that can only tolerate small deviations from the data-generating policy, this work provides a "first-principles derivation" that does not assume a specific training data distribution.

**Weaknesses:**

1. The analysis in sec 2.2 is novel and interesting, but as they stated in the paper, this deduction needs the surrogate objective in (11) holds. However, this objective normally does not hold in reality since the policy only takes gradient descent for each sample and won't achieve at the optimal point. Hence, the clear conditions of when the derivation will make one gradient descent improve should be rigorously proved as in TRPO [1].

Since TRPO provides rigorous proofs to discuss when importance sampling can make the algorithm converge, the work to disagree this approach should also prove convergence. Otherwise, the results are not convincing enough.

[1] John Schulman, et al, Trust Region Policy Optimization

2. The experimenters are not valid enough. In Figure 2, they only do experiments for 1.5b model and only one group of sync parameters. They are suggested to compare more base models and more different sync_interval and offset parameters to validate their results.

**Questions:**

See weaknesses.

---

> ### Author Response · Authors · 2025-11-22
> **Response to Reviewer 5sDW (1/2)**
>
> Thank you for your thoughtful and constructive review! We address your questions and concerns in the following.
>
> ---
>
> **Suggestion: experiments with more different settings.**
>
> Thank you for the suggestion!
>
> To strengthen the results of Figure 2 (in particular, its empirical validation for Finding F1), we have conducted additional experiments in a new setting that uses a mix of sync_interval = 16 and sync_offset = 8.
> The experiment results have been added to Figure 2 in our revision, which further confirm our Finding F1.
>
> We also want to clarify that Figure 2 is not the only empirical evidence for validating Finding F1.
> Indeed, the claim that "importance sampling might play a non-essential (i.e., not absolutely necessary) role" has been thoroughly validated across diverse settings:
>
> * Figure 2, 8, 9: Qwen2.5-1.5B-Instruct on GSM8k with various configurations of off-policyness and learning rates;
> * Figure 3: Llama-3.2-3B-Instruct on ToolAce;
> * Figure 5: Qwen2.5-7B-Instruct on Guru-Math, reward curves for GRPO and REC-OneSide-NoIS (with the same clipping range) are almost identical;
> * Figure 11: Llama-3.2-8B-Instruct on MATH, which again validates the efficacy of REC-OneSide-NoIS;
> * Figure 12: new experiments with Qwen3-30B-A3B (an MoE model) on Guru-Math; in response to Reviewer 5aHq's question about sequence-level clipping and importance sampling like in GSPO [1], we have implemented a REC-GSPO-NoIS variant that discards the importance-sampling weights in GSPO, and confirmed its efficacy empirically.
>
> We believe that such diversity of experiment settings and empirical evidence would be sufficient and convincing enough for validating our analytical results.
>
> [1] Group Sequence Policy Optimization. arXiv 2025.

---

> ### Author Response · Authors · 2025-11-22
> **Response to Reviewer 5sDW (2/2)**
>
> **Concerns about the analysis of Section 2.2.**
>
> Thank you for raising the questions. Let us make a few clarifications as follows.
>
> One concern by the reviewer is that, while "this deduction needs the surrogate objective in (11) holds", our analysis in Section 2.2 "only takes gradient descent for each sample and won't achieve at the optimal point".
> We believe that this doesn't constitute a real issue, since **the essential rationale behind our analysis does NOT really "need" (as stated in the reviewer's comment) the surrogate objective in Eq. (3) or (11) holds**.
> Instead, the surrogate objective serves merely as a guidance for the policy update direction, and different algorithms could adopt various ways of optimizing it approximately; this is analogous to the case of TRPO and PPO, which were built upon theoretical foundation developed in simplified settings, but also incorporate various approximations and heuristics for practical implementations and realistic settings.
> It is exactly *by design* that we only take one gradient descent step for the surrogate loss, which recovers a group-relative variant of classic REINFORCE while giving it a native off-policy interpretation.
> Solving the surrogate objective exactly, on the other hand, will recover the REBEL algorithm [1]; discussions about the similarities and differences between its derivation and our analysis in Section 2.2, as well as practical limitations of REBEL, can be found in Section 5 of our revised manuscript.
>
> Another concern by the reviewer is that, since the TRPO paper rigorously proves that importance sampling can make the algorithm converge (in simplified settings), "the work to disagree this approach should also prove convergence", whereas the current work --- as acknowledged in the "limitations" part of our manuscript --- provides no formal guarantee for convergence or policy improvement.
>
> + To clarify, **our work does NOT mean to "disagree this approach (of importance sampling)"**.
> Instead, we aim to provide new and complementary perspectives for understanding the working mechanisms behind various practical and popular algorithms (e.g., GRPO) in off-policy settings, through the unified lens of group-relative REINFORCE and its native off-policy interpretation developed in Section 2.2.
> Our analysis has been accompanied by thorough empirical results across diverse importance-sampling-free algorithms and controlled experiment settings,
> which support our claim that importance sampling could be non-essential for effective off-policy RL.
> It is worth noting that **throughout this work, "non-essential" literally means "not absolutely necessary"**, rather than "making no impact" or "useless/meaningless".
> Therefore, our findings do not contradict the rationale and efficacy of importance sampling in the conventional theory of policy gradient and TRPO.
>
> + Our focus in the current work is to present the most essential rationale for the proposed off-policy interpretation, its numerous algorithmic implications, and extensive empirical evidence that supports our analytical findings.
> Proving formal convergence guarantees --- while identifying weak and realistic assumptions about off-policy data distribution --- is another story.
> We anticipate that it could be fairly challenging, and might need to build upon a theoretical foundation that is substantially different from those (e.g., policy gradient theory) in prior literature.
> We thus refrain from attempting to give partial theoretical results, in order to avoid the risk of misleading others.
> Instead, we acknowledge that rigorous convergence guarantees are beyond the scope of our current focus, and hope that our work would inspire further exploration and research along this promising direction.
>
>
> [1] REBEL: Reinforcement Learning via Regressing Relative Rewards. NeurIPS, 2024.
>
> ---
>
> Thank you again for your review, which helps us a lot in improving the current work! We hope that our responses have provided satisfactory answers to your questions, but do let us know if you have any further concern that we can address.

---

> > ### Comment · Reviewer_5sDW · 2025-11-27
> >
> > Thank the authors for their concrete feedback! I have no more questions for the experiment.
> >
> > However, for the theory
> >
> > 1. The key point for their deduction is eqn (4), which is the closed-form solution of eqn (3), and this equation does not hold until (3) reaches the optimality. This is also the reason for the inequivalence of RLHF and DPO. Hence, I do not believe the claim that "the essential rationale behind our analysis does NOT really "need' (as stated in the reviewer's comment) the surrogate objective in Eq. (3) or (11) holds". The analysis needs more cautious claims.
> >
> > 2. I understand this work is not opposed to TRPO, but the off-policy title for GRPO is counterintuitive since if a policy is very far from the inference policy, the training must collapse. Of course the authors can try to remove the importanct ratio, but this must hold under some conditions, such as the policy must stay close to the reference policy. The authors should discuss more about that.
> >
> > Moreover, whether this algorithm can be extended to settings where staleness is larger is doubtful.

---

> > > ### Author Response · Authors · 2025-11-28
> > > **Response to Reviewer 5sDW**
> > >
> > > Thank you for your reply!
> > > We're glad to know that your initial questions about the experiments have been resolved.
> > > Let us clarify your remaining concerns about the theory in the following.
> > >
> > > ---
> > >
> > > **Recap: the main plot of this work.**
> > >
> > > + In Section 2.2, we propose a novel off-policy interpretation for group-relative REINFORCE.
> > > That is, we provide an alternative way of understanding what group-relative REINFORCE is *effectively* doing at the $t$-th iteration, as visualized in Figure 1: it defines the surrogate loss in Eq. (6) that incentivizes approaching the optimum of the surrogate objective in Eq. (3), but only takes one gradient descent step of the surrogate loss.
> > >
> > > + As clearly stated at the beginning of Section 3, the proposed off-policy interpretation itself doesn't guarantee convergence to the optimal policy when given arbitrary data (which is too good to be true).
> > > The main point of Section 3 is exactly to identify pitfalls that could undermine the convergence of vanilla group-relative REINFORCE, which further motivate two general principles for augmenting this algorithm: adding regularization and reshaping data distribution.
> > >
> > > + In Section 4, we reinterpret various existing LLM-RL algorithms as instantiations of these principles, which deepens our understanding of their working mechanisms. For example, our off-policy interpretation justifies removing importance-sampling weights (as done in REC-OneSide-NoIS) and discarding some samples strategically (as done in RED-Drop), even though they violate certain assumptions in the conventional theory for REINFORCE, such as unbiased policy gradient estimate. And these are not just theoretical; the efficacy of such algorithms has been validated by extensive empirical results.
> > >
> > >
> > > **Clarifications.**
> > >
> > > + In Section 2.2, what we propose is an off-policy interpretation for group-relative REINFORCE, not the algorithm itself.
> > >
> > > + Throughout this work, we have never intended to claim that group-relative REINFORCE (or any of its variants) converges to the optimal policy without additional assumptions about the data distribution.
> > >
> > >
> > > **Tentative revisions.**
> > >
> > > + Thank you for pointing out that "the off-policy title for GRPO is counterintuitive". In retrospect, we realize that the title is the only place where we say "group-relative REINFORCE is an off-policy algorithm", whereas in the manuscript, we consistently state "group-relative REINFORCE admits an off-policy interpretation". **We're open to revising the title** to "Group-Relative REINFORCE Secretly Admits an Off-Policy Interpretation: [...]" (or something similar), if the reviewer finds this more accurate and reasonable than the original title.
> > >
> > > + Regarding the content of this manuscript, we have made our best efforts in ensuring the clarity of our writing, which has been acknowledged by other reviewers as well. Nonetheless, it would be greatly appreciated if the reviewer could identify specific parts of this manuscript that could mislead or confuse the readers, and we'll be happy to revise them accordingly.
> > >
> > >
> > > **Reviewer's comment: "if a policy is very far from the inference policy, the training must collapse."**
> > >
> > > This is not necessarily true.
> > > For example, consider a didactic two-arm bandit setting with a softmax policy. By a simple analysis (even simpler than the one presented in Section 3), we can prove that group-relative REINFORCE converges to the optimal policy (i.e., probability 1 for the arm with the highest reward), even if samples are drawn from a fixed inference policy.
> > >
> > >
> > > **Reviewer's summary of this work: "This paper argues that popular reinforcement learning (RL) algorithms for LLMs, like Group Relative Policy Optimization (GRPO), are *mistakenly* treated as on-policy methods"**
> > >
> > > This is a misunderstanding of what we intend to claim.
> > > Our work aims to provide a complementary off-policy interpretation for group-relative REINFORCE and GRPO (which could potentially broaden their range of applicability), rather than to refute their classical on-policy interpretation.
> > >
> > >
> > > ---
> > >
> > > Thank you again for your valuable feedback! We hope that our responses have addressed your concerns regarding the theory in this work.

---

### Official Review · Reviewer_vzAn · 2025-11-05

**Soundness:** 2
**Presentation:** 4
**Contribution:** 2
**Rating:** 4
**Confidence:** 4

**Summary:**

The authors frame a GRPO variant as an off-policy algorithm, and use it to justify and unify several recent empirical heuristics under different themes.

**Strengths:**

(+) The exposition of this paper was crystal-clear, well done! I appreciated the structuring and how clearly key results were called out.

(+) It feels like every day on arXiv, we see a new GRPO variant. I appreciate the author's efforts to bring some clarity to this fast-moving space.

**Weaknesses:**

(-) A couple citations are missing. I'd suggest adding in one for MaxEnt IRL above Eq. 4, and to https://arxiv.org/abs/2406.01462 on line 176 re: coverage conditions for the unique minimizer to be the soft-optimal policy. Up to rescaling, I believe the square loss (Eq. 6) is the same as the loss function defined in https://arxiv.org/abs/2404.16767, so I would also add in a reference to that. I might also suggest adding in a reference to https://arxiv.org/abs/2505.20686 in Sec. 4.2.

(-) I find the characterization of REINFORCE as "inherently off-policy" a bit strange. I believe what Sec. 2.2 actually proved is that given off-policy data, performing a single step of square-loss minimization leads to group-relative REINFORCE. Whether this data was generated on- or off-policy is an entirely orthogonal point as far as I can see. Let me know if I've misunderstood things.

(-) It would be good to clarify early on in this paper that group-relative REINFORCE and GRPO refer to different things -- I believe the former doesn't include the standard deviation term in the advantage estimate?

(-) I'm not sure how the theory implies that the square loss penalty in Sec. 4.2 is "principled." Similarly, I don't understand how the theory implies that randomly up-weighting samples is "well-justified."

**Questions:**

(1) In https://arxiv.org/abs/2404.16767, authors analyze a similar square-loss objective and note that taking a single-step of Gauss Newton recovers the Natural Policy Gradient. They also discuss the variance-reduction benefit of using other samples in the same "group" as a baseline. Could you comment on the similarities and differences between your work and theirs? Also, I believe the REBEL algorithm in that paper is precisely computing the $\tilde{\theta}s$ in Figure 1 -- could you confirm this?

(2) Can you spell out the point of Sec. 4.1's experiments more explicitly? Also, re: AsymRE, it might be good to explicitly write out how this shift effects the closed-form solution -- I think it should boil down to a change in $\tau$.

---

> ### Author Response · Authors · 2025-11-22
> **Response to Reviewer vzAn (1/3)**
>
> Thank you for your thoughtful and constructive review! We address your questions and concerns in the following.
>
> ---
>
> **Question: citations and discussions for some related works.**
>
> Thank you for referring us to these prior works [1, 2, 3, 4]. They are indeed quite relevant and helpful; we have cited them in the revised manuscript according to your suggestions.
>
> We elaborate on some **similarities and differences between our work and the REBEL paper [1]** --- which have been incorporated into the revised manuscript --- as follows:
>
> + Similarities:
> Steps 1 and 2 of our derivation in Section 2.2 mostly overlap with the derivation of REBEL in [1]; these include KL-regularized reward maximization as the surrogate objective in Eq. (3), its closed-form solution in Eq. (4), the pairwise consistency condition in Eq. (5), and the squared surrogate loss in Eq. (6).
> We certainly do not claim novelty over these results, which have long been standard and well-known facts.
> Indeed, as discussed in "related works" (Section 5), CoPG [5] is another closely related algorithm that follows similar steps in its derivation (except that it uses a fixed reference policy $\pi_{\text{ref}}$ for KL regularization, rather than $\pi_{\theta_{t}}$).
>
> + Differences:
>     + In Step 3 of our derivation in Section 2.2, we take a single gradient step of the squared loss, which leads to a group-relative variant of REINFORCE.
>     In contrast, REBEL proposes to solve the squared loss at each iteration $t$, or equivalently, to compute $\tilde{\theta}_{t+1}$ in Figure 1.
>
>     + The above difference has implications on infrastructure complexity and efficiency of the algorithms.
>     Many enterprise‑grade LLM-RL frameworks, such as verl and Trinity-RFT, have been highly optimized for REINFORCE-style algorithms (including all those investigated in our work).
>     For example, in the presence of data parallelism and gradient accumulation, these frameworks can automatically divide a mini-batch into multiple micro-batches (each containing multiple or just one sequence) in a way that maximizes load balancing and training efficiency, while minimizing peak memory usage.
>     However, solving the squared loss directly (like REBEL or CoPG does) contradicts these performance optimization techniques, as it requires paired responses for the same prompt to be located within the same micro-batch.
>     This constraint increases infra complexity and peak memory usage, as reported in [3].
>
>     + One interesting observation in [1] is that applying Gauss-Newton to the squared loss will recover Natural Policy Gradient (NPG).
>     We view this as a different way of approximately optimizing the squared loss, which doesn't appear to have a direct connection to our Step 3 analysis in Section 2.2.
>
>     + The NPG formula in [1] contains "reward - baseline" terms that are analogous to those in RLOO, for the special case of two responses per prompt.
>     Through derivation and rationale that are quite different from those in [1], our Step 3 analysis in Section 2.2 arrives at group-relative REINFORCE that involves the $\\{r_i - \bar{r}\\}_{1 \le i \le K}$ terms, for the general case of $K > 1$ responses per prompt.
>
>     + The REBEL paper considers a one-step contextual bandit setting, where the LLM generates a response $y$ when given a prompt $x$.
>     In comparison, our core analysis in Section 2.2 has been extended to a multi-step RL setting where each rollout trajectory contains multiple turns of agent-environment interaction, as presented in Appendix A.
>
>
> [1] REBEL: Reinforcement Learning via Regressing Relative Rewards (arxiv:2404.16767)
>
> [2] The Importance of Online Data: Understanding Preference Fine-tuning via Coverage (arxiv:2406.01462)
>
> [3] Accelerating RL for LLM Reasoning with Optimal Advantage Regression (arxiv:2505.20686)
>
> [4] Maximum entropy inverse reinforcement learning (AAAI 2008)
>
> [5] Contrastive Policy Gradient: Aligning LLMs on sequence-level scores in a supervised-friendly fashion (arxiv:2406.19185)

---

> ### Author Response · Authors · 2025-11-22
> **Response to Reviewer vzAn (2/3)**
>
> **Question: "I find the characterization of REINFORCE as 'inherently off-policy' a bit strange."**
>
> Thank you for pointing this out.
> We agree that the original title of Section 2.2 might be a bit confusing, and we have changed it from "A new interpretation: REINFORCE is inherently off-policy" to "A new off-policy interpretation for group-relative REINFORCE" in the revision.
>
> To further clarify: throughout this work,
> * By saying "Algorithm X is an on-policy algorithm", we mean that the essential rationale behind X requires on-policy samples;
> * By saying "Algorithm X is an off-policy algorithm" or "X admits an off-policy interpretation", we mean that the essential rationale behind X makes no explicit assumption about the data distribution, thus allowing both on- and off-policy samples.
>
> ---
>
> **Suggestion: "It would be good to clarify early on in this paper that group-relative REINFORCE and GRPO refer to different things -- I believe the former doesn't include the standard deviation term in the advantage estimate?"**
>
> Thank you for the suggestion.
> Indeed, "group-relative REINFORCE" --- as presented in Line 62 - 76 --- refers to a group-relative variant of the classical REINFORCE algorithm, using group mean reward as the baseline for advantage calculation.
> It doesn't include other elements of GRPO, such as advantage normalization by the standard deviation of rewards.
>
> To clarify this distinction early on in the paper, we have revised the abstract by adding an one-sentence description of group-relative REINFORCE.
>
> ---
>
> **Questions: "I'm not sure how the theory implies that the square loss penalty in Sec. 4.2 is 'principled.' Similarly, I don't understand how the theory implies that randomly up-weighting samples is 'well-justified.'"**
>
> Regarding the first question, we'd like to clarify that the main point of our study about Kimi's OPMD in Section 4.2 is not about the squared-loss penalty itself, but rather about its overall loss $\hat{L}$.
> It is a sum of two loss terms: vanilla group-relative REINFORCE (which, as we have shown, admits a native off-policy interpretation), and a squared-loss penalty that incentivizes keeping $\pi_{\theta}$ close to $\pi_{\text{old}}$.
> We say this is "principled" because it perfectly fits into our proposed framework --- in particular, the "regularize policy update step" principle presented in Section 3 --- without the need for various approximations required in the original derivation of OPMD in the Kimi k1.5 paper.
>
> Regarding the second question, we'd like to clarify that we haven't claim that *randomly* up-weighting samples is "well-justified".
> Instead, for the RED-Weight method presented in Section 4.3, we consider up-weighting *high-reward* samples, as explained in Line 455.
> This is justified by decomposing the policy update as in Line 458 - 462, which ends up being the sum of (1) pairwise-weighted group-relative REINFORCE as in Eq. (9), and (2) a regularization term that incentivizes imitating higher-reward samples.
> It is crucial that the regularization term has a *positive* coefficient $\bar{r}_{w} - \bar{r} > 0$,
> precisely because we up-weight samples with higher rewards (rather than randomly).

---

> ### Author Response · Authors · 2025-11-22
> **Response to Reviewer vzAn (3/3)**
>
> **Question: "Can you spell out the point of Sec. 4.1's experiments more explicitly?"**
>
> Thank you for raising this question, which indicates room for improvement in our writing.
>
> We have revised the "experiments" paragraph in Line 351 - 368 by explicitly spelling out (at the beginning) the main point of these experiments,
> which is to validate Finding F1 --- the roles of clipping and importance sampling in GRPO, and the efficacy of enlarging the clipping range ---  presented in the table of page 6.
>
> ---
>
> **Question: "re: AsymRE, it might be good to explicitly write out how this shift effects the closed-form solution -- I think it should boil down to a change in $\tau$."**
>
> We'd appreciate some clarifications about this question: by saying "closed-form solution", do you mean that of the KL-regularized surrogate objective in Eq. (3)?
> If the answer is yes, then we must apologize for the confusion: the symbol $\tau$ in Section 4.2 of our original submission represents the coefficient of a penalty term added to the vanilla REINFORCE loss (cf. Line 377), which is *different from* the parameter $\tau$ in Eq. (3) and Section 2.2.
>
> To avoid such confusion, we have replaced $\tau$ with $\beta$ in Section 4.2 of our revised manuscript.
>
> ---
>
> Thank you again for your review, which helps us a lot in improving the current work! We hope that our responses have provided satisfactory answers to your questions, but do let us know if you have any further concern that we can address.

---

### Author Response · Authors · 2025-12-03
**General response to AC, SAC and PC (1/2)**

Dear AC, SAC and PC,

We sincerely appreciate your efforts in meta-reviewing our work!

We understand that the peer-review process for ICLR this year has been particularly challenging.
To assist this process, we provide in this letter a summary of the reviews that we received, responses and revisions that we made, and follow-up reviewer-author discussions.

---

**Strengths and contributions of this work.**

We appreciate that **all reviewers** acknowledged the core contributions of our work:

1. We have presented novel and foundational analysis for a group-relative variant of classic REINFORCE,
showing that it admits a native off-policy interpretation without explicit assumption about the data distribution
(whereas the conventional theory of REINFORCE requires on-policy samples for unbiased policy gradient estimate);
2. Through this unified lens, we reinterpret various LLM-RL algorithms (GRPO, OPMD, AsymRE, distribution-reshaping heuristics, etc.) as augmented variants of group-relative REINFORCE, offering new insights and clarity to understanding their working mechanisms in off-policy scenarios.

Our key results and messages have been delivered by clear writing (Reviewers vzAn, ZsF7) and validated by extensive experiments (Reviewer ZsF7).

One particular comment from Reviewer vzAn says, *"It feels like every day on arXiv, we see a new GRPO variant. I appreciate the author's efforts to bring some clarity to this fast-moving space."*
We deeply appreciate the reviewer's recognition of our efforts,
and hope that our work could make valuable and timely contributions to the LLM-RL community in this way.


---

**Our responses and revisions.**

During the rebuttal period, we have made our best efforts to address every question and concern raised by the reviewers:

+ Reviewer 5aHq was curious whether our Finding 1 (regarding the working mechanisms of GRPO) can generalize to sequence-level clipping and importance sampling, like in GSPO. We have conducted additional experiments and provided a positive answer.
+ As suggested by Reviewer 5sDW, we have conducted additional experiments for a new off-policy setting, which further enhance the empirical validation of our main results.
+ We have added citations and discussions for some related works, as requested by Reviewers vzAn and ZsF7.
+ Reviewer 5sDW raised concern about the necessity of formal convergence guarantees. This concern mostly arises from the reviewer's misunderstandings about our intended objective in this work and misconceptions about the behavior of group-relative REINFORCE or GRPO in off-policy scenarios, and we have clarified them in our response.
+ We have addressed writing and presentation issues, and clarified other initial misunderstandings or confusion by the reviewers.

We appreciate the constructive suggestions from the reviewers, which have helped us a lot in improving this work.
Although follow-up reviewer-author discussions have been limited (partly because the discussion phase was closed early),
**we believe that our responses and revisions have resolved almost all concern and questions raised by the reviewers**,
with one exception that is explained in the following.

---

> ### Author Response · Authors · 2025-12-03
> **General response to AC, SAC and PC (2/2)**
>
> **The remaining misunderstanding by Reviewer ZsF7: the focus, contributions and values of this work.**
>
>
> In her/his initial review, Reviewer ZsF7 commented that *"If the authors are unable to design more effective and efficient algorithms based on their proposed theory, then the contribution of this work appears to be overstated"*,
> but without specifying what overstatements precisely were made in our manuscript.
> In our response, we clarified the novel algorithmic ingredients introduced in this work, and explained the values of new insights and better understanding for foundational algorithms like GRPO (which include inspiring future algorithmic developments).
>
> Reviewer ZsF7 replied by reiterating *"I believe the contribution is below the acceptance threshold unless the authors can develop more effective algorithms based on their theory."*
> **We respectfully disagree with such a claim.**
> As far as we understand, **ICLR has been an inclusive community that welcomes research papers of diverse types**,
> including papers that focus on developing novel algorithms and achieving SOTA scores, as well as papers that focus on advancing understanding, providing theoretical insights, or reinterpreting existing methods in ways that facilitate future progress.
> This can be seen from the official [ICLR Reviewer Guide](https://iclr.cc/Conferences/2026/ReviewerGuide),
> which provides broad definitions of originality, significance, contributions, values and interest of a research work,
> as detailed in [our follow-up response to the reviewer](https://openreview.net/forum?id=7CFlXvCoN6&noteId=elj9mNmXFX).
>
>
>
> ---
>
>
> We hope that you will find the above information useful when making the final evaluation and decision about our work.
> Thank you again for your time and efforts in ensuring a constructive and fair review process!
>
>
> Best,
>
> Authors

---

### Meta-Review · Area_Chair_6ejY · 2026-01-07

**Summary:**

The reviewers were split about this paper and did not come to a consensus. On one hand they appreciated the clarity of the paper and the theoretical framework introduced by the paper. On the other they had issues with (a) the characterization of REINFORCE as “inherently off-policy”, (b) how the theory implies that a squared loss penalty is principled, (c) missing citations, (d) unjustified approximations of the optimal policy, (e) missing experiments, (f) lack of a novel algorithm, (g) related work discussion, (h) lack of detailed analysis about the clipping rate. The authors responded to concerns (a)-(c) convincingly. The authors argue that complaint (d) applies equally to TRPO and PPO. This doesn’t exactly address the complaint but contextualizes it: the goal of the paper is not to improve such approximations but theoretically relate GRPO variants, and use this relationship to derive a naive off-policy algorithm. I consider this point resolved. The authors run additional experiments, resolving complaint (e). A novel algorithm is not required for acceptance to ICLR, complaint (f) is resolved. The authors expand the related work section, addressing complaint (g). The authors add additional context on the clipping rate, and admit that their result does not have theoretical backing, but is a surprising finding. This is far from the main contribution of the paper so I consider this resolved. Based on this, the authors have responded to all reviewer concerns. For this reason, I vote to accept.

**Reviewer Concerns:**

Please see meta-review.

**Reviewer Scores:**

I believe the reviewers would have kept or increased their scores.

---

### Decision · Program_Chairs · 2026-01-26

Accept (Poster)